# Shuttle peptide delivers base editor RNPs to rhesus monkey airway epithelial cells in vivo

Katarina Kulhankova [1], Soumba Traore[1], Xue Cheng[2], Hadrien Benk-Fortin[2], Stéphanie Hallée[2], Mario Harvey[2], Joannie Roberge[2], Frédéric Couture[3], Sajeev Kohli[4,5,6], Thomas J. Gross [7], David K. Meyerholz [8], Garrett R. Rettig [9], Bernice Thommandru[9], Gavin Kurgan [9], Christine Wohlford-Lenane[1], Dennis J. Hartigan-O'Connor[10,11], Bradley P. Yates[12], Gregory A. Newby[4,5,6,12], David R. Liu[4,5,6], Alice F. Tarantal[11,13,14], David Guay [2] & Paul B. McCray Jr. [1] ✉

Gene editing strategies for cystic fibrosis are challenged by the complex barrier properties of airway epithelia. We previously reported that the amphiphilic S10 shuttle peptide non-covalently combined with CRISPR-associated (Cas) ribonucleoprotein (RNP) enabled editing of human and mouse airway epithelial cells. Here, we derive the S315 peptide as an improvement over S10 in delivering base editor RNP. Following intratracheal aerosol delivery of Cy5-labeled peptide in rhesus macaques, we confirm delivery throughout the respiratory tract. Subsequently, we target *CCR5* with co-administration of ABE8e-Cas9 RNP and S315. We achieve editing efficiencies of up-to 5.3% in rhesus airway epithelia. Moreover, we document persistence of edited epithelia for up to 12 months in mice. Finally, delivery of ABE8e-Cas9 targeting the *CFTR* R553X mutation restores anion channel function in cultured human airway epithelia. These results demonstrate the therapeutic potential of base editor delivery with S315 to functionally correct the *CFTR* R553X mutation in respiratory epithelia.

Gene editing offers the opportunity to repair or modify mutations associated with inherited diseases such as cystic fibrosis (CF), a disorder caused by mutations in CF transmembrane conductance regulator *(CFTR)*. While CFTR small molecule modulator therapies are available for certain mutations, the products of *CFTR* premature termination codon (PTC) alleles are not responsive. Base editing using adenine deaminase proteins provides an opportunity to repair PTCs by efficiently installing single base changes with minimal undesired byproducts.

CRISPR-based adenine base editors (ABEs) use catalytically inactive or nickase Cas variants fused to an evolved deoxyadenosine deaminase protein to specifically target point mutations by converting an A•T to G•C[1]. Notably, these RNA-guided programmable editors catalyze site specific single nucleotide conversions without DNA double-strand breaks. Base editing relies on ubiquitously expressed cellular mismatch repair machinery and does not require cell division[2]. Thus, the base editing process may proceed efficiently in post-mitotic cells in vivo, overcoming a

[1]Department of Pediatrics, University of Iowa, Iowa City, IA, USA. [2]Feldan Therapeutics, Quebec, Canada. [3]TransBIOTech, Lévis, Quebec, Canada. [4]Merkin Institute of Transformative Technologies in Healthcare, Broad Institute of MIT and Harvard, Cambridge, MA, USA. [5]Department of Chemistry and Chemical Biology, Harvard University, Cambridge, MA, USA. [6]Howard Hughes Medical Institute, Harvard University, Cambridge, MA, USA. [7]Department of Internal Medicine, University of Iowa, Iowa City, IA, USA. [8]Department of Pathology, University of Iowa, Iowa City, IA, USA. [9]Integrated DNA Technologies, Coralville, IA, USA. [10]Department of Medical Microbiology and Immunology, School of Medicine, UC Davis, Davis, CA, USA. [11]California National Primate Research Center, UC Davis, Davis, CA, USA. [12]Department of Genetic Medicine, Johns Hopkins University School of Medicine, Baltimore, MD, USA. [13]Department of Pediatrics, School of Medicine, UC Davis, Davis, CA, USA. [14]Department of Cell Biology and Human Anatomy, School of Medicine, UC Davis, Davis, CA, USA. ✉e-mail: paul-mccray@uiowa.edu

limitation in the airways for strategies requiring homologous recombination[3,4].

For disorders involving the respiratory tract such as CF, a critical challenge is the delivery of the gene editing payload to the epithelial cells of the conducting airways. We and others recently demonstrated the feasibility of using CRISPR-based ABEs to correct *CFTR* PTC mutations when delivered to cultured airway epithelial cells as RNPs via electroporation[5] or as mRNA[6]. Several delivery strategies including viral and non-viral vectors are in development to enable the delivery of editing reagents to somatic cells in vivo, including the lung[7–12]. One challenging but attractive approach is to deliver ABEs as an RNP complex. In contrast to coding RNA or DNA, RNP delivery allows immediate and efficient editing while the transient cell exposure limits the off-target opportunities. While Yeh et al. successfully delivered Cas9-BE3 (CBE) RNPs to edit inner ear cells of mice using cationic lipids[2], we are unaware of an effective lipid-based reagent for RNP delivery to airway epithelia[13]. We previously reported successful CRISPR-associated nuclease delivery to the respiratory tract of mice using an engineered amphiphilic S10 shuttle peptide[13]. We achieved editing of loxP sites in airway epithelia of ROSA^mT/mG mice, offering potential avenues for nuclease and base editor RNP delivery to refractory airway epithelial cells in vivo.

Here, we identify an optimized shuttle peptide and prepare recombinant ABE8e-Cas9 RNP to investigate shuttle-mediated delivery and base editing efficiency in relevant airway epithelial cells. To our knowledge, we demonstrate for the first time the feasibility of base editing of airway epithelia in the rhesus monkey model. In addition to in vivo base editing in rhesus monkeys, using the Ai9 ROSA26 tdTomato reporter mouse model we document the persistence of in vivo edited airway epithelial cells for 12 months. Importantly, the efficiency of editing with this delivery strategy is sufficient to restore CFTR function in cultured primary CF airway epithelial cells.

## Results

### Shuttle peptide with improved delivery of ABE RNPs to airway cells

We previously demonstrated the successful delivery of green fluorescent protein (GFP), Cas12a and Cas9 RNPs to airway epithelia in vitro and in vivo using the amphiphilic S10 peptide[13]. From a peptide screen in well differentiated primary cultures of human airway epithelia grown at an air-liquid interface we identified four additional peptides (termed S321, S195, S262, and S315) that delivered Cas9 RNP more efficiently than S10 (Fig. 1a and Supplementary Table 1). Based on this result, we asked whether the improved Cas9 RNP delivery to human cells by S315 could also drive more efficient gene editing in airway epithelia.

The primary amino acid sequences of the S315 peptide were derived from the S10 sequence and feature a similar N-terminal hydrophobic cluster and C-terminal hydrophilic/cationic tail, and an identical poly-glycine linker (Fig. 1b). Note that the S315 peptide has a lower cationic charge density (+8) than S10 (+10). Both basic residue clusters found in S10 (KK/RR highlighted by red squares) were each reduced to a single residue in the S315 peptide (black squares). To reduce the hydrophobicity of the C-terminus, the single leucine in the S10 C-terminus was moved to the N-terminus of S315 (arrow). Finally, for uniformity, the S315 peptide contains only leucines (L) as highly hydrophobic residues, lysines (K) as basic residues, and glutamines (Q) or alanines (A) as uncharged residues.

We previously found that S10-mediated delivery of Cas12a RNP with its shorter gRNA was more efficient than S10 delivery of Cas9 RNP. This may be due to inhibitory effects that the greater anionic charge density of the longer Cas9 gRNA has on peptide-mediated delivery[13]. As S315 presents a lower overall charge (+8) than S10 (+10) and lacks both KK and RR cationic clusters, we investigated the sensitivity of S10 and S315 peptides to inhibition of delivery by the polyanionic charges associated with gRNAs used with Cas9 RNPs (Fig. 1c and

Supplementary Fig. 1). To determine the relative activity maintained in the presence of Cas9 RNP, we applied GFP to the CFF-16HBEge human bronchial epithelial cell line in the presence or absence of Cas9 RNPs and quantified GFP delivery by S10 or S315 using flow cytometry as previously described[13]. The inclusion of Cas9 RNPs markedly reduced the GFP delivery efficiency using S10, while addition of Cas9 RNPs had little impact on S315 peptide-mediated delivery which retained ~85% of its delivery activity.

### Adenine base editing in human and rhesus monkey airway cells in vitro

As Cas9 and ABE8e-Cas9 share the same sgRNA, we next compared the utility of S10 and the more efficient S315 peptide for ABE8e-Cas9 RNP delivery to the CFF-16HBEge cells grown in submersion culture[14]. The frequency of ABE8e-Cas9 base editing at the targeted *B2M* locus was quantified by high-throughput DNA sequencing (HTS). The *B2M* locus was selected because it is a ubiquitously expressed gene product in diverse cell types including airway epithelia. The editing efficiency for ABE8e-Cas9 RNP delivered by the S315 peptide was ~20%, a significant increase over the S10 peptide (Fig. 2b). To investigate the feasibility of adenine base editing in the rhesus monkey model, we targeted the safe harbor *CCR5* locus, which we could assess in vivo in future experiments. We delivered ABE8e-Cas9 RNPs targeting the *CCR5* locus to primary cultures of rhesus monkey tracheal epithelial cells grown at an air-liquid interface using the S10 or S315 peptide. As shown in Fig. 2d, the editing attained using the S315 peptide was greater than with S10, achieving a mean editing efficiency of ~9%.

### Amphiphilic peptides enhance delivery of a protein cargo in vivo

To evaluate shuttle peptide delivery biodistribution in vivo, we designed a nucleus-targeted peptide cargo labeled with a Cy5 dye. To generate this cargo, we synthesized a peptide comprised of a nuclear localization signal (NLS). We used D-amino acids to avoid protease degradation and a retro-inverso sequence to conserve the amino acid residue disposition of the NLS[15]. Finally, we chemically conjugated a sulfo-Cy5 fluorophore to a cysteine residue at the peptide C-terminal end to generate the NLS-Cy5 cargo as described in Methods (Production of DRI-NLS-Cy5). When co-administered with a shuttle peptide, the nuclear Cy5 signal signifies successful delivery and avoidance of the endosomal entrapment often associated with cell penetrating peptides. In these experiments we used the S10 peptide since its GFP delivery activity in the absence of Cas9 RNP was greater than that observed with S315 (Supplementary Fig. 1). As outlined in the Methods (Rhesus procedures), we delivered 1 ml of NLS-Cy5 (10 μM) formulated with the S10 peptide (40 μM) into the trachea of young rhesus monkeys using an atomizer. Between one and two hours post-delivery animals were euthanized, and lung lobes collected, fixed, and prepared for examination (Supplementary Table 2, Group 1).

Using epifluorescence (Fig. 3a) and confocal microscopy (Fig. 3b) we documented NLS-Cy5 signal in surface epithelial cells of the large and small conducting airways, and the alveolar regions. We observed Cy5 signal in ciliated cells (acetylated tubulin+) and secretory cells (SCGB1A1+) of the large cartilaginous airways and in smaller airways lined by a cuboidal epithelium (Fig. 3b, top panel). To assess delivery to cells with progenitor capacity we focused on airway basal cells. In the pseudostratified columnar epithelium of the conducting airways the basal progenitor cells are located below the differentiated surface cell types. In the small airways typified by a cuboidal epithelium, the basal progenitor cells have an apical membrane that is accessible from the lumen[16]. Sporadic co-localization of NLS-Cy5 with the basal cell marker cytokeratin 5 (CK5) was observed (Fig. 3b, middle panels).

In the lung parenchyma, we identified cells with NLS-Cy5 positive nuclei that co-localized with surfactant protein C (SP-C), a marker of alveolar type II cells (Fig. 3b, bottom panels). Alveolar type II cells are a progenitor cell in the lung parenchyma. Some NLS-Cy5 positive cells

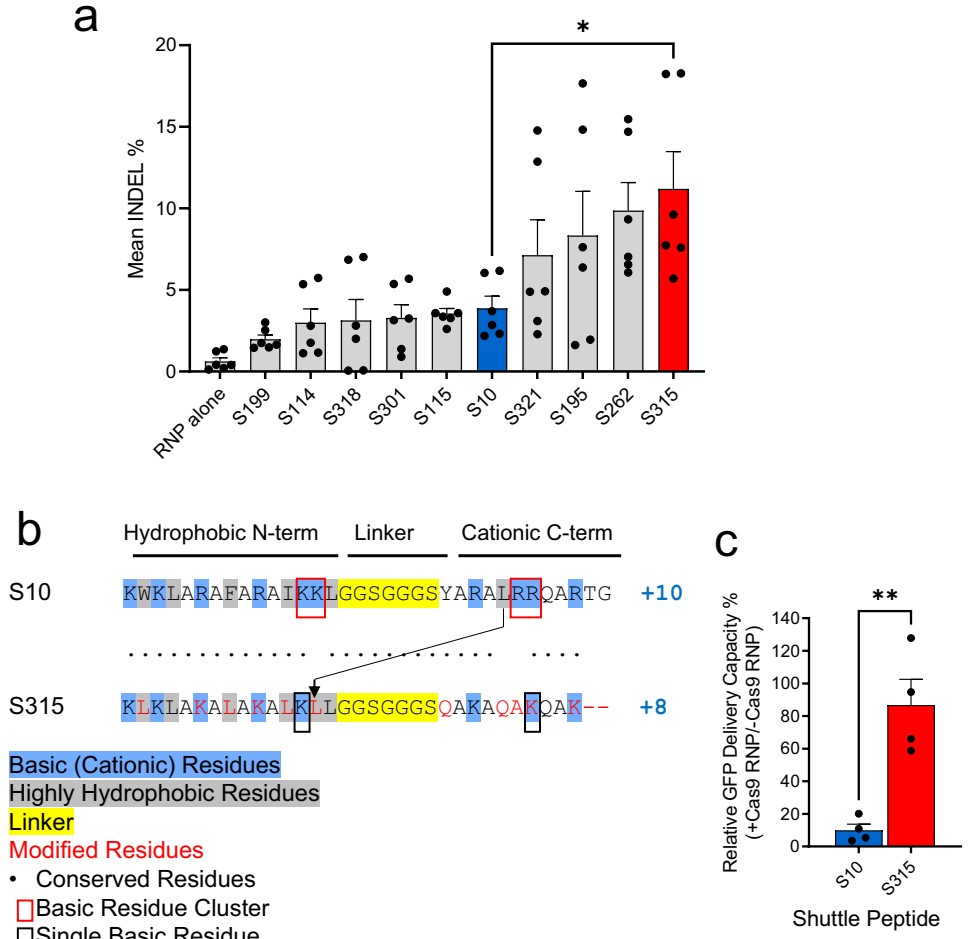

**Fig. 1 | Identification of shuttle peptides with improved delivery of Cas9 RNPs to airway epithelia. a** Delivery of Cas9 RNP (ribonucleoprotein) targeting *CFTR* locus in human airway epithelial cells cultured at the air liquid interface using indicated shuttle peptide candidates. Y axis represents the frequency of indels (insertions, deletions, and substitutions in the quantification window) attained with the indicated peptide. Individual closed circles represent data from $n = 6$ independent samples. Blue and red color represent the peptides used further in the study (S10 and S315, respectively). Results plotted as mean + SEM. Statistics by unpaired two-tailed *t* test, *$P = 0.01$. **b** Comparison of the amino acid sequences of S10 and S315 peptides. **c** Inhibitory effect of Cas9 RNP on S10- or S315-mediated delivery of GFP to CFF-16HBEge cells. GFP (green fluorescent protein, 10 μM), S10 or S315 (10 μM) peptide with or without Cas9 RNP (containing 2.5 μM Cas9 and 2 μM gRNA) were added to cells and GFP delivery quantified by flow cytometry. The Y axis represents the relative delivery activity (%), calculated as the GFP delivery attained with or without Cas9 RNP addition. Results plotted as mean + SEM. Statistics by unpaired two-tailed *t* test, **$P = 0.003$. Individual circles represent average data from $n = 20,000$ cells examined over 4 independent experiments. Source data are provided as a Source Data File.

were also observed in the alveolar lumen that co-localized with CD68, a marker for alveolar macrophages. In macrophages the Cy5 signal was cytoplasmic, suggesting ingestion of free NLS-Cy5 peptide, rather than transduction via shuttle peptide delivery (Fig. 3b). This result provides further support that shuttle peptides designed to specifically deliver proteins to the cell cytoplasm facilitate efficient import of the NLS-tagged cargo to the nuclei of airway epithelial cells. The observed heterogeneity in the percentage of cells targeted regionally presumably reflects variable delivery of materials from a single intratracheal aerosol bolus. Figure 4a shows a schematic of rhesus airway anatomy. Figure 4b presents the regional distribution of the NLS-Cy5 peptide signal in trachea, bronchus, and large and small airways of the five lung lobes sampled; the bars are color coded to correspond with Fig. 4a. The range of NLS-Cy5 labeled cells was 0.5–20.8% in the large airways and 1.0%–17.8% in the small airways.

### Editing of the CCR5 locus in rhesus monkey airway epithelia in vivo

Using an approach identical to that outlined for the NLS-Cy5 peptide, we delivered 1 ml of instillation solution containing ABE8e-Cas9 RNP

(2.5 μM Cas/2 μM gRNA final concentrations; 40 μM Cas/100 μM crRNA and tracrRNA stock concentrations) formulated with either the S10 or S315 peptide (40 μM final concentration; 250 μM stock concentration) by intratracheal aerosol as described in Methods (Rhesus procedures). All animals were prescreened for study selection and those negative for antibodies to *Sp*Cas9 assigned to the study. They were monitored daily and showed normal activities and food intake. Complete blood counts (CBCs) and clinical chemistry panels prior to and post-administration were all within normal limits. Circulating inflammatory cytokines were assessed at the pre- and post-administration time points by cytokine bead array including IL-6, IL-10, CXCL10 (IP-10), IL-1β, IL-12p40, IL-17A, IFN-β, IL-23, TNF-α, IFN-γ, GM-CSF, CXCL8 (IL-8), and CCL2 (MCP-1). No statistically significant differences between the pre- and post-administration blood samples were detected. Seven days post-administration the final blood samples were collected, animals were euthanized, lung lobes collected, airway epithelia obtained using cytology brushes, and DNA extracted and subjected to HTS (Supplementary Table 2, Group 2).

The airway regions sampled by cytology brushings are shown schematically in Fig. 4a. The editing efficiencies in the trachea, left and

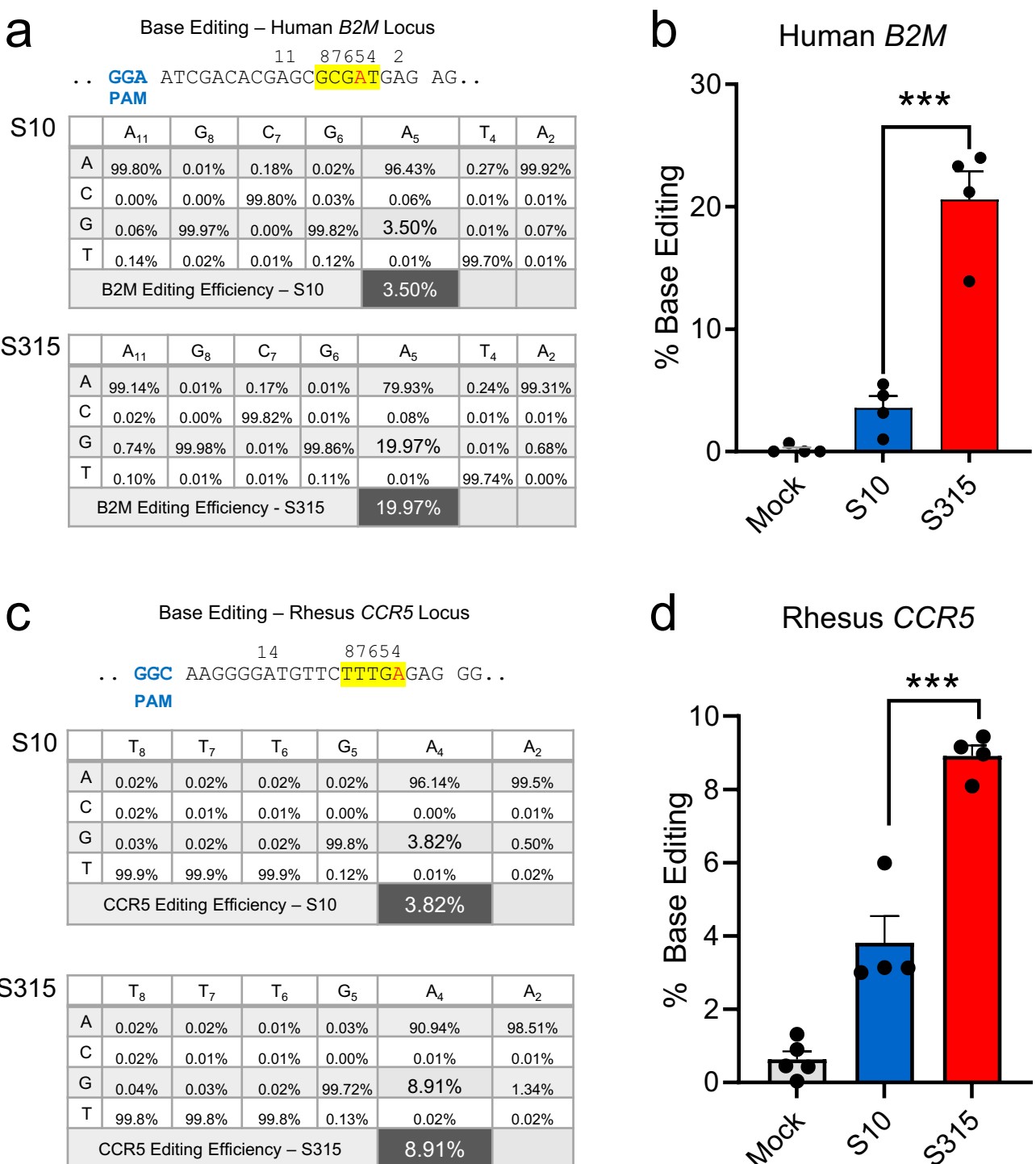

**Fig. 2 | Adenine base editing in human and rhesus monkey airway cells in vitro.** **a** Delivery of ABE8e-Cas9 RNP targeting *B2M* locus using S10 and S315 peptides in human airway epithelial cells cultured at the air liquid interface. Editing efficiency was assessed using HTS. Top panel shows the target DNA strands and PAM sites (blue font). Yellow highlight denotes the predicted 4–8 nt ABE editing window (numbered). Target A residue is in red font. Average frequency of desired product for human *B2M* locus with the indicated shuttle peptides. **b** Graphic representation of desired base editing efficiency at human *B2M* locus using S10 (blue column) and S315 (red column) peptides. Individual closed circles represent data from 4 technical replicates, and the data is plotted as mean ± SEM. Statistics by unpaired two-tailed *t* test, \*\*\**P* = 0.0005. **c** Delivery of ABE8e-Cas9 RNP targeting *CCR5* locus using

S10 and S315 peptides in rhesus tracheal epithelial cells cultured at the air liquid interface. Editing efficiency quantified using HTS. Top panel shows the target DNA strands and PAM sites (blue text). Yellow highlight denotes the predicted 4–8 nt ABE editing window (numbered). Target A residue highlighted in red text. Target A residue values are inlarge font. Average frequency of desired product for rhesus *CCR5* locus with the indicated shuttle peptides. **d** Graphic representation of desired base editing efficiency at rhesus *CCR5* locus using S10 (blue) and S315 peptides (red). Individual closed circles represent data from 4 technical replicates, and the data are plotted as mean ± SEM. Statistics by unpaired two-tailed *t* test, \*\*\**P* = 0.0006. Source data are provided as a Source Data File.

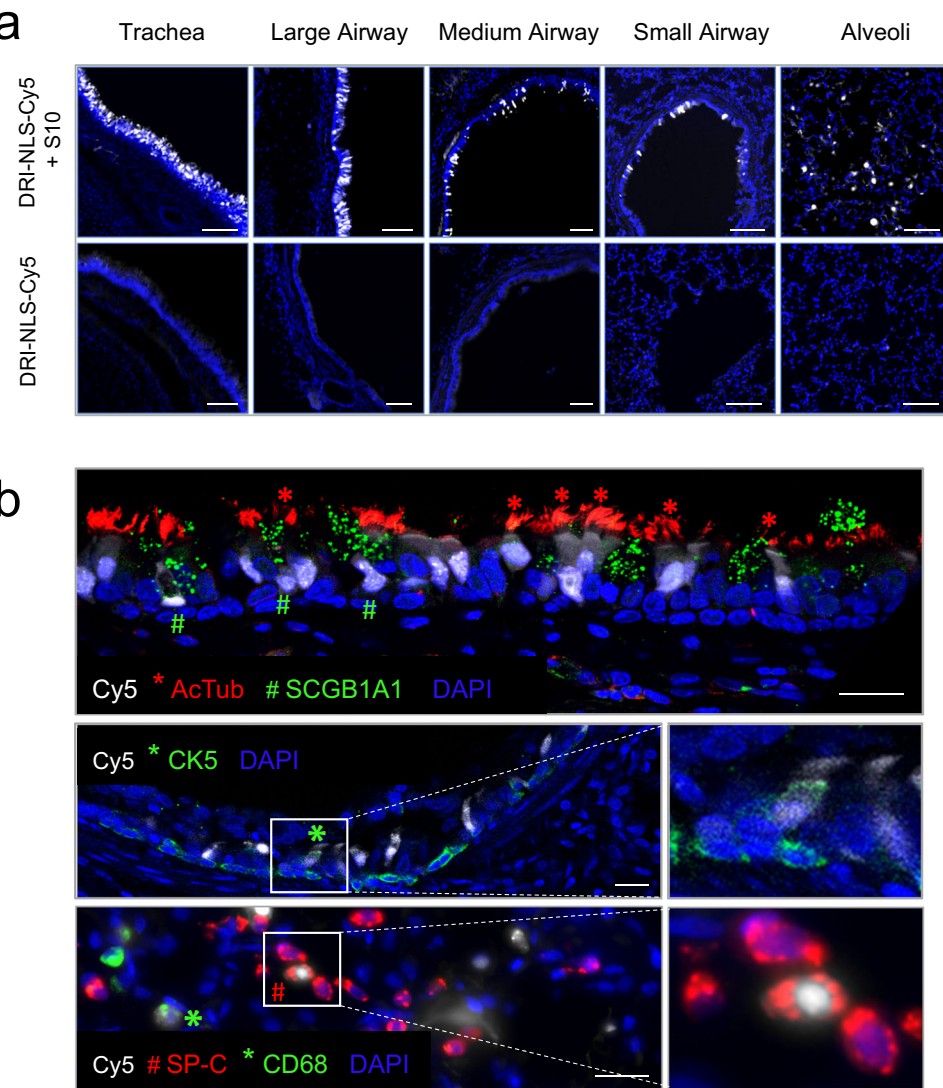

**Fig. 3 | Intratracheal in vivo delivery of DRI-NLS-Cy5 with S10 shuttle peptide results in widespread distribution and cell-type specific delivery in rhesus monkey airways. a** Representative epifluorescence microscopy images of the rhesus lung tissue sections demonstrating the DRI-NLS-Cy5 localization in the surface epithelial cells of airways of various sizes, ranging from trachea to small airways and alveolar regions (Fig. 3a, top panel). Control received DRI-NLS-Cy5 alone (Fig. 3a, bottom panel). Scale bar is 100 μm. **b** Confocal microscopy images documenting localization of DRI-NLS-Cy5 fluorescence (white) in ciliated cells (*, AcTub - red), secretory cells (#, SCGB1A1 – green, top panel), and very rarely to CK5+ basal cells (*, CK5 – green, middle panel), where DAPI is pseudo-colored blue. In the alveolar regions (Fig. 3b, lower panel), the DRI-NLS-Cy5 signal (white) localized to the nuclei of the surfactant protein C producing alveolar type II cells (#, SP-C - red) and alveolar macrophages (*, CD68 - green). Scale bar is 20 μm. For (**a**) and (**b**), the staining and microscopy was performed at least two times on tissue sections from different regions of the lungs and yielded similar results.

right mainstem bronchi, and segmental bronchi are presented in Fig. 4c. Controls included animals that received NLS-Cy5 alone, NLS-Cy5 + S10 shuttle, and ABE8e-Cas9 RNP with no shuttle (Supplementary Table 2, animals 1–4, 5). Little editing was observed at the *CCR5* locus with the ABE8e-Cas9 RNP alone or S10 peptide delivery of the ABE8e-Cas9. In contrast, in the two animals that received the S315 shuttle peptide with ABE8e-Cas9 RNP we measured a mean base editing efficiency of 2.8% (range 0.2–5.3%) depending on the airway region sampled (Fig. 4c). The most efficient editing was observed in the left upper lobe caudal part (5%), left upper lobe cranial part (3.7%), and the right mainstem bronchus (3.8%) in monkey #7 and in the trachea (5.3%), right lower lobe (5.1%) and left lower lobe (4.6%) in monkey #8, both of which received the S315 shuttle peptide.

We assessed off-target base editing following the in vivo delivery of *CCR5*-targeted ABE8e RNPs into macaques using S315 peptide. We first used CIRCLE-seq[17] to identify top candidate off-target sites in the rhesus monkey genome that can be recognized by *CCR5*-targeted Cas9

RNP complexes. We designed primers to PCR-amplify the top 14 of these regions from the rhesus monkey genome for high-throughput sequencing to assess editing frequencies. Nine of these candidate off-target sites amplified well and were readily sequenced (Supplementary Tables 4, 5). No off-target editing was detected at any of these nine sites above the background of sequencing error (Supplementary Fig. 2, Supplementary Table 6). We anticipate that the transient duration of exposure to RNP editing reagents used in this study relative to viral vectors or even mRNA delivery methods will generally minimize off-target editing, as previously reported[18].

We obtained computed tomography (CT) scans for all animals pre- and immediately post-intratracheal aerosol delivery of all cargoes. We anticipated heterogenous deposition of editing materials within the airway tree following a single 1 ml aerosol administration. As an example, Fig. 5a displays the pre- and post-delivery chest CT images from monkey #7 (S315 + RNP). We hypothesized that changes in aeration post-delivery might serve as a surrogate for regions receiving the

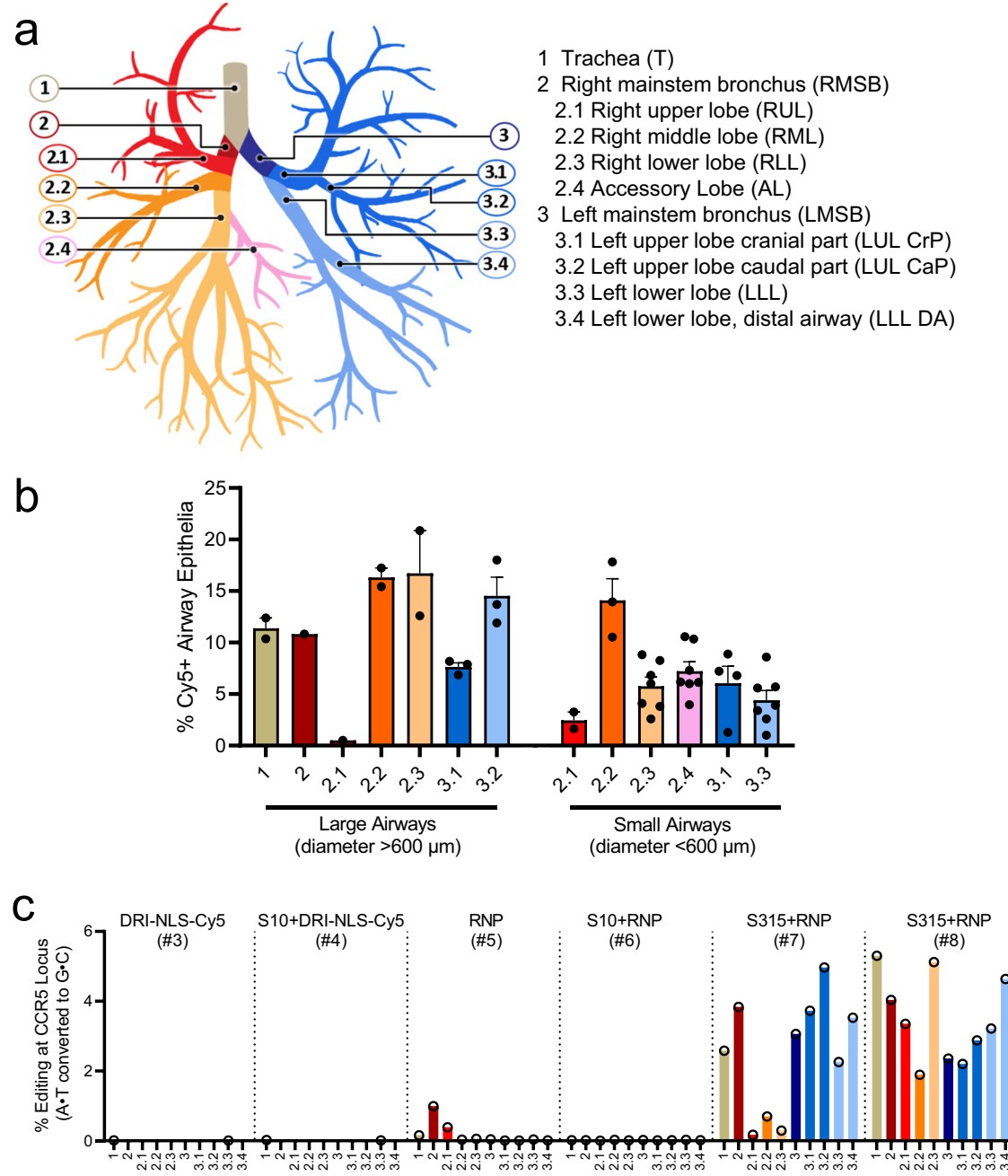

**Fig. 4 | Quantification of in vivo delivery of DRI-NLS-Cy5 and ABE8e-Cas9 RNP to rhesus respiratory epithelia. a** Diagram of rhesus monkey airway tree. The regions where airway tissue sections or cytology brushings were obtained are color coded and numbered as indicated. **b** Quantification of DRI-NLS-Cy5 delivery with S10 peptide in trachea (T), right mainstem bronchus (RMSB), and 7 lobar anatomical locations (RUL - right upper lobe, RML - right middle lobe, RLL - right lower lobe, AL - accessory lobe, LUL-CrP - left upper lobe, cranial part, LUL CaP - left upper lobe, caudal part, LLL - left lower lobe). Each circle represents one airway analyzed in a single tissue section from a given anatomical location, and columns represent mean ± SEM. Between 1 and 7 airways per section were analyzed, $n = 1$ animal. **c** Efficiency of shuttle peptide mediated Cas9-ABE8e RNP editing at *CCR5* locus scored by airway region. Y axis indicates A to G editing efficiency. X axis denotes conditions including DRI-NLS-Cy5 alone (#3), S10 + DRI-NLS-Cy5 (#4), ABE8e-Cas9 RNP alone (#5), S10 + ABE8e-Cas9 RNP (#6), and S315 + ABE8e-Cas9 RNP (#7, 8). Each condition presents data from an individual animal. The animal numbers correspond to conditions described in Supplementary Table 2 ($n = 6$ animals). Source data are provided as a Source Data File.

greatest aerosol deposition. A pulmonologist blinded to the experimental conditions scored the CT scans for the presence and extent of parenchymal lung changes indicated by areas of abnormal density. CT scans were examined in multiple planar reconstructions and the parenchymal abnormalities then assigned to corresponding proximal conducting airways per the anatomic schema in Fig. 4a. A four-grade scale (ranging from no change (NC) to +++) was used to characterize

new findings in individual lung lobes after cargo delivery. The results are presented in Fig. 5b. Indeed, the areas with the greatest findings characterized and scored as nonsegmental or segmental parenchymal density changes or consolidation correlated with greater *CCR5* locus editing (monkeys #7, 8) or Cy5+ cells (monkey #4), suggesting greater reagent deposition in those regions. As noted above, CBCs and clinical chemistry panels immediately pre-delivery and prior to euthanasia

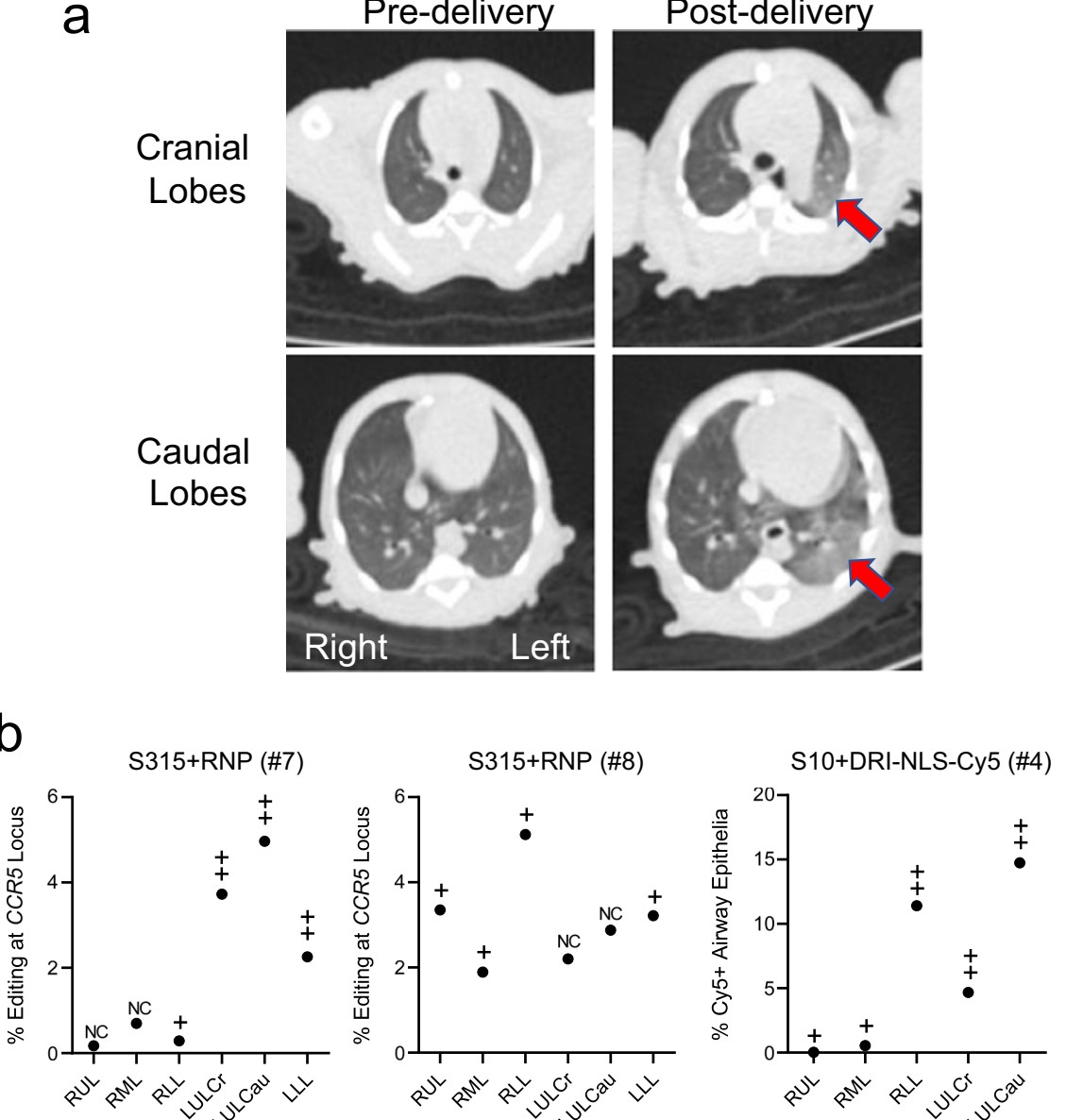

**Fig. 5 | Application of chest CT scans to identify regions of deposited base editing reagents. a** Chest CT from monkey #7 (S315 + RNP) from (**b**) below. Arrows highlight areas of consolidation. **b** Correlation between regional editing efficiency or regional DRI-NLS-Cy5 nuclear localization and areas of consolidation on CT scan. Regions studied include RUL - right upper lobe, RML - right middle lobe, RLL - right lower lobe, LUL-Cr - left upper lobe, cranial part, LUL Ca - left upper lobe, caudal part, LLL - left lower lobe. The CT scans were scored in a blinded fashion for changes in aeration as follows: NC: no change from baseline; + subtle nonsegmental consolidation; ++ segmental consolidation; +++ dense consolidation. Filled circles represent editing efficiencies for indicated region as shown in Fig. 4c. Source data are provided as a Source Data File.

were all within normal limits for control, NLS-Cy5, and ABE8e-Cas9 treated animals.

## In vivo persistence of editing
Our data demonstrate that shuttle peptide delivery mainly targets the accessible surface ciliated and secretory cells in the airways (Fig. 3). To investigate the long-term persistence of gene edited airway epithelial cells in vivo, we used the Ai9 ROSA26 tdTomato mouse model[19]. The main surface epithelial cell types of the murine airways are ciliated and secretory. In these animals, nuclease mediated excision of a LoxP flanked stop codon in the *Rosa26* locus activates tdTomato expression (Supplementary Fig. 3a). Using methods similar to a previous report[13], the MAD7 nuclease[20] RNP was delivered intranasally using the S10 peptide. MAD7 is an engineered Cas12a variant type isolated from the

bacterium *Eubacterium rectale*. At intervals of 1 week and 3, 6, and 12 months, lung tissues were collected and tdTomato expression in epithelial cells evaluated and quantified using fluorescence microscopy as described in Methods (Microscopy and quantitative analysis of mTomato+ cells) (Supplementary Fig. 3b, c). We observed persistence of tdTomato expressing cells over 12 months with an average of 8.3% edited airway epithelial cells across all time points (Supplementary Fig. 3c). There was a statistically significant decline in the number of tdTomato⁺ cells from 7 days to 12 months.

## In vivo pulmonary toxicity study in mice
We previously investigated the toxicity profile of S10 and CRISPR nuclease RNPs in mice[13]. To better understand the lung response to peptide mediated ABE8e-Cas9 RNP delivery, we delivered shuttle

peptides S10 or 315 alone or in combination with the RNP via intranasal instillation. At 1 and 7 days post-delivery we performed broncho-alveolar lavage (BAL), total and differential cell counts, lung histo-pathology analysis, and profiled cytokine and chemokine mRNA transcript abundance by RT qPCR. These results are presented in Supplementary Figs. 4–6. We observed a transient mild cellular immune response and low-grade inflammation at 1 day post instillation that was resolving at 7 days post instillation. There were minimal changes in the abundance of any transcript, and no sustained increases in the peptide + ABE8e-Cas9 RNP groups.

### ABE RNP delivery to CF airway epithelia partially restores CFTR function

We next asked if ABE8e-Cas9 RNP delivery with the S315 peptide could restore function to cells with a *CFTR* nonsense mutation. We performed these studies in primary human airway epithelial cells heterozygous for the R553X mutation (R553X/L671X). We produced and purified an ABE8e-Cas9 NG protein for this study as no NGG PAMs are available at the human *R553X* locus[21] (Fig. 6a). ABE8e-Cas9 RNP was delivered to well differentiated air-liquid interface cultures as described in Methods[22], section Primary cultures of human and rhesus monkey airway epithelia. One week following ABE8e-Cas9 delivery, editing was quantified by HTS and CFTR-dependent Cl⁻ secretion was measured. The allelic editing efficiency achieved with S315 delivery (4.92%) exceeded that observed with the S10 peptide (2.36%) (Fig. 6b, c). To evaluate the functional impact of the A•T to G•C editing, we measured CFTR-dependent anion channel activity in Ussing chambers. Epithelia were sequentially treated with amiloride to inhibit epithelial sodium channels (ENaC) and DIDS to inhibit non-CFTR Cl⁻ channels. We next applied forskolin and IBMX (F&I, cAMP agonists) to activate CFTR-dependent Cl⁻ secretion (measured as change in short circuit current, $\Delta I_{sc}$, Fig. 6d). Activation of CFTR was assessed by the addition of the CFTR channel inhibitor GlyH-101 (GlyH, Fig. 6e). We observed significant increases in CFTR-dependent Cl⁻ transport following ABE8e-Cas9 delivery with S10 and S315 peptides (Fig. 6d, e). Representative tracings from these experiments are shown in Fig. 6f. The CFTR-dependent short circuit current observed following S315 mediated delivery was greater than that of S10, consistent with the HTS results.

## Discussion

Here we report, to the best of our knowledge, the first demonstration of the translational potential of shuttle peptides for protein and ABE8e-Cas9 RNP delivery to respiratory epithelia in the rhesus monkey model. Following a single aerosol administration, we successfully delivered a fluorescently labeled protein cargo to the epithelial cells of large and small airways, and to some alveolar epithelia. Using the S315 shuttle for ABE8e-Cas9 RNP delivery, we attained significant A to G editing of the *CCR5* locus in cells recovered using bronchial brushing. The editing efficiency of the *CCR5* site in epithelia harvested from the trachea and proximal airways reached 5.3%. While this editing efficiency is at the low end of the therapeutic range, application of this delivery approach in human CF airway epithelia with the R553X mutation achieved similar levels of editing and conferred partial restoration of CFTR function.

We previously reported the feasibility of CRISPR nuclease RNP delivery to human airway epithelial cells in vitro and the airway epithelia of mice in vivo using shuttle peptides[13]. The present demonstration of successful adenine base editing in rhesus airway epithelia provides further support for translational protein cargo delivery using the versatile shuttle peptide technology. In these studies, the S315 shuttle provided more efficient ABE8e-Cas9 RNP delivery than the S10 shuttle used previously[13]. This likely reflects peptide modifications that included a decreased overall charge density and/or the disruption of cationic charge clusters that impact interactions with the negatively

charged gRNA used with Cas9. Despite a lower GFP delivery activity on CFF-16HBEge cells, these modifications allowed the S315 peptide to maintain its activity in presence of Cas9 RNP (Fig. 1c) which may explain its superiority in Cas9 and ABE8e-Cas9 RNP delivery to well differentiated human airway epithelial and rhesus tracheal epithelial cells in vitro (Fig. 2a, b) and to rhesus airway epithelia in vivo (Fig. 4c).

There is currently a lack of consensus regarding the % of cells that must be corrected to restore CFTR function. To achieve non-CF levels of Cl⁻ transport, it is estimated that CFTR function should be restored in 5–50% of the airway epithelial cells[23–28]. We note that CF patients with mutations associated with as little as 10% residual CFTR function may have mild disease phenotypes, including little or no lung disease[29]. Importantly, not all cell types participate equally in Cl⁻ secretion. Several recent scRNA-seq studies have elucidated a diversity of cell types in the large airway surface epithelium (e.g., basal, secretory, goblet, club, ciliated, ionocyte, neuroendocrine, hillock)[30–32]. While ionocytes express the highest levels of *CFTR* transcripts, they are a rare cell type and their function is still a subject of study, as is their role in genetic therapies for CF[30,31,33,34]. One notable finding from these studies is that *CFTR* transcript abundance varies greatly among individual cell types. Okuda and colleagues used scRNA-seq, single cell RT PCR, and scRNA in situ hybridization to demonstrate that secretory cells are the dominant airway surface cell type for CFTR expression and function[32]. In contrast, ciliated surface cells exhibited low and infrequent CFTR expression[32]. In the same study, secretory cells comprised ~15% of the epithelium. Single cell RNA-seq studies of airway epithelia demonstrated that secretory cells express the Na-K-2Cl cotransporter-1 (NKCC1 or SLC12A2), a basolateral membrane Cl⁻ entry pathway required for Cl⁻ secretion[30–32]. We speculate that because airway epithelial cells are electrically coupled by gap junctions[35–37] Cl⁻ may move between secretory cells with no functional CFTR to those that are corrected by gene editing. Thus, any cell with new functional CFTR channels is poised to support Cl⁻ secretion and may provide a conduit through its connections to neighboring cells. Our results in human airway epithelia with the R553X mutation indicate that A to G editing of ~5% of surface epithelia partially restored CFTR function. This suggests that restoring CFTR function in a small proportion of surface epithelial cells may be beneficial. More studies are needed in CF models to better understand the contributions of individual epithelial cell types to CFTR function and to identify the preferred cell types to target with gene editing and gene therapy strategies.

The use of CF animal models that develop pulmonary manifestations may also provide valuable information on shuttle activity in diseased lung tissue. CF patients accumulate abnormal mucus in the airways enriched with mucins, inflammatory cells, proteases, DNA, and bacteria which could interfere with delivery[38,39]. Preliminary studies indicate that shuttle peptide delivery activity can be maintained in the presence of CF sputum by adding protease inhibitors or using a D-amino acid shuttle peptide (Supplementary Fig. 7a). Since using D-amino acids to synthesize the shuttle does not affect delivery activity in vitro and in vivo (Supplementary Fig. 7a, b), it represents a simple approach to translate this technology to lung diseases such as CF.

Several factors may influence the outcome of ABE8e-Cas9 RNP delivery including the preferentially edited cell types, the amount of RNP delivered, the gRNA affinity for its target, and the accessibility of the RNP to target DNA. While *CCR5* is not an abundant transcript in airway epithelia[40], it has been studied extensively as a genomic safe harbor site for gene therapies[41], and its low-level expression suggests accessible open chromatin. For successful Cas9 RNP delivery, we believe that the quantity of delivered cargo may not be limiting. Rather, the shuttle peptides that achieve delivery to the greatest percentage of cells are expected to exhibit the most efficient editing, independent of the quantity of material delivered.

The study presented here has strengths and limitations. We demonstrated base editing and characterized immune responses

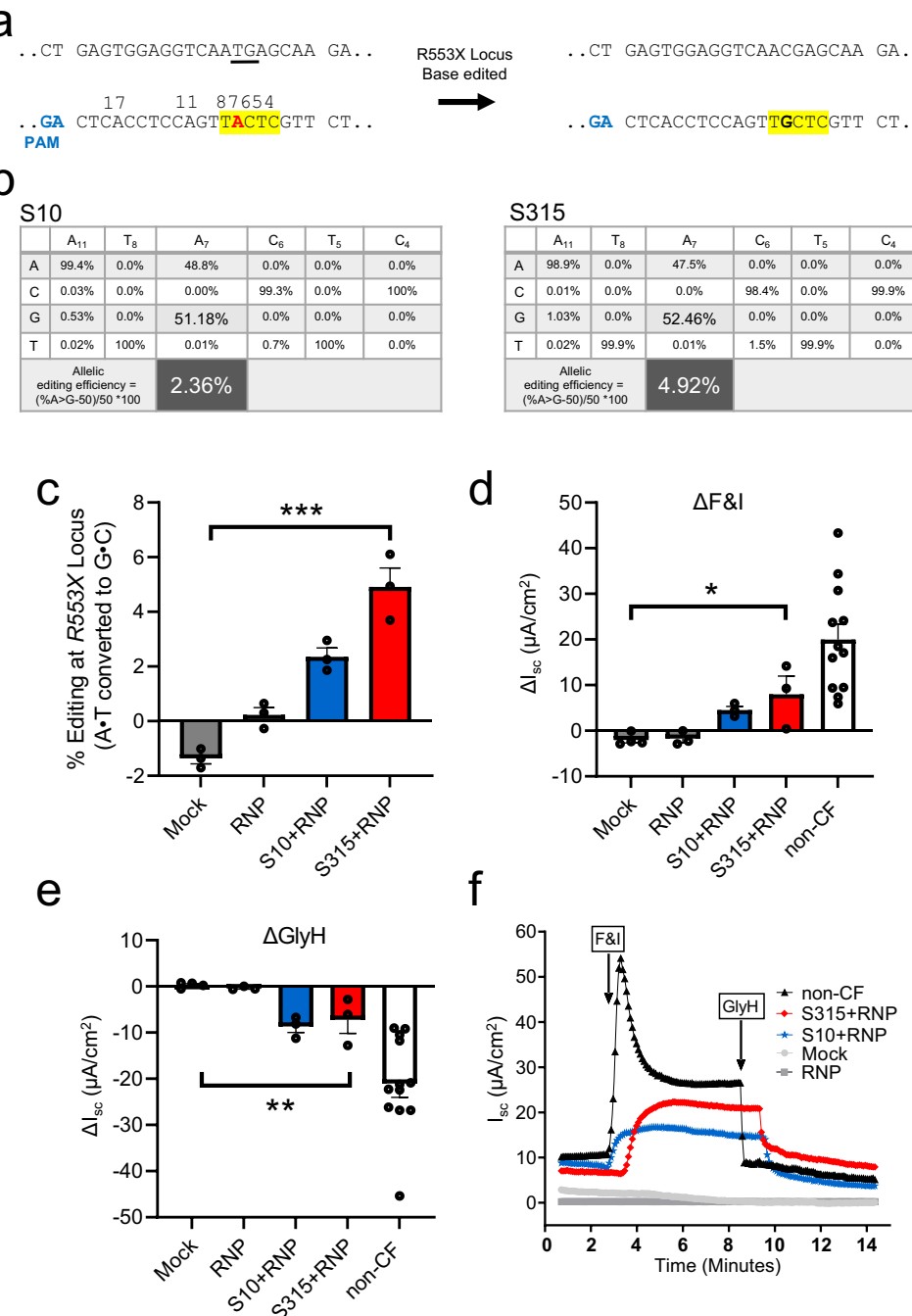

**Fig. 6 | Shuttle peptide delivery of ABE8e-Cas9 RNP to primary air liquid interface cultures of human airway epithelial cells (*R553X/L671X*) targeting *R553X* locus.** One week following the first application of ABE8e-Cas9 RNP delivery with shuttle peptides, Ussing chamber analysis was conducted, and DNA editing was analyzed by HTS. **a** Top panel shows the target DNA strands and PAM sites (blue text). Yellow highlight denotes the predicted 4–8 nt ABE editing window (numbered). Mutations highlighted in red text. **b** Average frequency of desired product and allelic editing efficiencies for R553X nonsense mutation with the indicated shuttle peptides. Percent allelic editing efficiencies calculated by (% base edited-50)/50 *100 and graphically represented in (**c**) and the values are highlighted inlarge font. Note that the cells were compound heterozygous for the *CFTR* R553X mutation. Thus, all cells contain "wild type" sequence for one allele and have the R553X mutation on the other allele. This mathematical adjustment provides a

meaningful "per cell" estimate of correction. Statistics by ordinary one-way ANOVA without any adjustments, ***$P < 0.0001$. **d, e** CFTR-dependent anion channel activity summarized from short circuit current tracings across all treatment groups. Change in short circuit current ($\Delta I_{sc}$) in response to F&I (forskolin & IBMX) (**d**) and GlyH (**e**) in groups represented in (**c**) and non-CF donor cells. Statistics by ordinary one-way ANOVA without any adjustments, *$P = 0.01$, **$P = 0.003$. **f** Representative short circuit current tracings comparing mock, ABE8e-Cas9 RNP alone ABE8e-Cas9 RNP + S10 (blue), and ABE8e-Cas9 RNP + S315 (red) treated cells. For (**c**–**e**), the CF human airway epithelia are represented by 3 technical replicates (treatment goups) or 4 (Mock group) technical replicates, and non-CF human airway epithelia are represented by 12 biological replicates. Each data point represents one cell culture. Data are plotted as mean ± SEM. Source data are provided as a Source Data File.

following non-viral delivery to the young rhesus monkey model, which has airway anatomy and physiology that closely represents humans. We also established that screening of peptides in the air-liquid interface culture model of well-differentiated human airway epithelia successfully identified shuttle peptides with improved in vivo delivery properties. The delivery of base editors as RNPs offers advantages as the duration of exposure to the editing agent is short and ends as the protein is degraded. This is expected to limit immunogenicity and off-target editing. Peptide and base editor proteins should also be amenable to rapid Good Manufacturing Practices production and chemistry, manufacturing, and control testing. Gene editing was achieved without evidence of toxicity in rhesus monkeys during the study period. Additional pulmonary toxicity studies in mice demonstrated mild cellular inflammatory changes that were resolving by day 7 post instillation.

Limitations of the in vivo rhesus monkey study include the small sample size for each experimental condition and the short-term (one-week) monitoring. Although 2–5% genome editing could be sufficient to confer some therapeutic benefit in humans, we would ideally aim for greater editing efficiency to further improve patient outcomes. Future goals include the evaluation of reagent formulation options, the testing of alternative delivery devices at multiple time points, and a longer study period in the primate model. We note that a single intratracheal aerosol delivery resulted in heterogeneous deposition of the reagents. A future goal will be to consider ways to optimize more uniform delivery. We suspect that repeat administration of the peptide and cargo may increase the number of edited airway surface cells but could result in immune responses and adverse events. It is possible that edited surface cell types may persist for long time periods thus it remains important to assess the long-term outcomes of single administration. Our results in Ai9 mice following editing using MAD7 nuclease editing demonstrated the persistence of edited cells in vivo over a 1-year period with some decrease at the last time point (Supplementary Fig. 3c). Previous studies in mice demonstrated that ciliated cells of the large and small airways are long-lived (half-life of 6 months in the trachea and 17 months in bronchioles[42]). Detailed information regarding airway cell turnover in humans is unavailable. Our results suggest that redosing of base editing RNPs in the airways could be necessary at intervals of 1 year or greater.

We observed that NLS-Cy5 delivery was more efficient than ABE8e-Cas9 RNP delivery. There are possibly several reasons for this finding. The NLS-Cy5 is a synthetic peptide of 24 amino acids (VKRKKKPPAAHQSDATAEDDSSYC) with an estimated molecular weight of 2.6 kDa. In contrast, the ABE8e-Cas9 is a much larger protein of 1614 amino acids and molecular weight of 185 kDa. We previously observed that a smaller cargo (FITC-Dextran 10 kDa) was more efficiently delivered by an amphiphilic peptide than a larger cargo (FITC-Dextran 250 kDa)[43]. Moreover, used as RNP, the guide RNA further increases the size and brings negatively charged nucleic acid moieties to the ABE8e-Cas9 protein which could reduce shuttle peptide delivery activity, albeit this impacts S315 less than the S10 shuttle. Additionally, our assessment of successful delivery varies between the two cargos. While the efficiency of NLS-Cy5 delivered cells is measured directly by visualizing fluorescent epithelial cells, the ABE delivery efficiency is indirectly measured by the gene editing efficiency. Since the editing efficiency is influenced by the individual guide RNA and the structural stability of the large RNP, it is possible that not all cells that successfully received ABE RNPs go on to have measurable editing.

Editing of accessible progenitor cell types is expected to result in permanent correction of the targeted cell and its progeny. There are several regional progenitor cell types in the conducting airways[44] including basal cells (Muc5AC⁻, CK5⁺, p63⁺) in the proximal cartilage containing tracheobronchial epithelium[45,46], club cells, and a population of basal and α6β4⁺ cells in the small airways[47,48]. Of note, there is some evidence that basal cells in the proximal airways have cell

membrane extensions that reach the lumen of the airway[49], while the progenitor cells of the small airways (basal, club, and α6β4⁺ cells) are directly accessible from the lumen[47,48]. Furthermore, there may be conditions in which secretory cells serve as progenitor cell types[50,51].

In summary, a single shuttle-peptide mediated delivery of ABE8e-Cas9 RNPs to the airways of young rhesus monkeys achieved up to 5.3% editing efficiency in airway epithelia. This finding supports the feasibility of attaining clinically relevant levels of gene editing using this protein delivery technology.

## Methods
### Ethical statement
All studies involving human cells and animal models were approved by the appropriate institutional agencies as outlined in the specific sections below.

### Production of DRI-NLS-Cy5
DRI-NLS-Cy5 was obtained by conjugating DRI-NLS-Cysteine peptide with Sulfo-Cy5 as described below. DRI-NLS-cysteine peptide (VKRKKKPPAAHQSDATAEDDSSYC-NH2, D-isoform) was synthesized by Expeptise (Saint-Laurent, Quebec, Canada). The Sulfo-Cy5-Maleimide was purchased from Lumiprobe (Hunt Valley, Maryland, USA). The DRI-NLS-cysteine peptide was dissolved in H2O and mixed with two equivalents of Sulfo-cy5-Malemide dissolved in acetonitrile. The conjugation reaction was performed at room temperature, monitored by UPLC and purified by HPLC.

### Expression and purification of recombinant ABE8e and MAD7
SpCas9 nuclease proteins (catalog # A037-a-0500PMOL, Feldan Therapeutics, Quebec, QC), were provided by Feldan Therapeutics. Recombinant ABE8e-Cas9 protein was purified from bacteria. The expression vector ABE8e-Cas9/pD881-SR (kanamycin resistant) was provided by Dr. David Liu. The E. coli BL21(DE3)pLysS strain (ThermoFisher) was freshly transformed with the Cas8abe8e/pD881-SR plasmid, and positive bacteria clone was grown at 37 °C until reaching an OD600 of 1.5. The culture was then incubated at 17 °C for one hour before adding 0.8% Rhamnose to induce protein expression. Bacteria were grown for another 20–24 h at 17 °C and finally harvested by centrifugation. Pellets were frozen at −80 °C until protein purification.

The bacterial pellet was thawed and resuspended in lysis buffer (40 mM sodium phosphate, 200 mM NaCl, 20 mM imidazole, 10 mM MgCl₂, 0.3% Triton X-100, 10% glycerol; pH 7.5) + 1 mM PMSF, 0.2 mg/ml lysozyme and 15 μg/ml DNAse I. The cell suspension was incubated at 4 °C for 1 h with agitation. The cell lysate was then clarified by centrifugation at 12,500 g, 10 °C, 40 min, following by a filtration step onto a 0.45 μm PES filter.

All purification steps were performed on an AKTApure 150 FPLC system (Cytiva). First, the lysate was passed through a HisPrep FF 16/10 (Cytiva) and washed with 20 volumes of 4% Heparin buffer B (40 mM sodium phosphate, 500 mM NaCl, 500 mM imidazole, 10% glycerol; pH 7.5), and the ABE8e-Cas9-containing fraction eluted with 70% of buffer B. The eluted sample was diluted in 1 volume of Heparin buffer A (40 mM sodium phosphate, 10% glycerol; pH7.2), and loaded onto 2 × 5 ml Hitrap heparin HP columns previously equilibrated with 20% buffer B (40 mM sodium phosphate, 1 M NaCl, 10% glycerol; pH 7.2). The ABE8e-Cas9 was eluted with a linear gradient from 20 to 100%B. Eluted fractions were pooled and concentrated using AMICON 100 K (MilliporeSigma) before loading onto a Superdex 200 pg 26/600 column in the storage buffer (25 mM HEPES, 350 mM NaCl, 10% glycerol, 1 mM DTT; pH 7.5). The peak corresponding to ABE8e-Cas9 protein dimers was concentrated on AMICON 100 K to a final concentration of 40 μM, sterilized by filtration 0.2 μm, aliquoted under the laminar flow hood, snap-frozen in liquid nitrogen, and stored at −80 °C. The final product had an endotoxin concentration of <0.250 EU/μg. The MAD7 nuclease was prepared as described previously[20].

 

## Amphiphilic peptide production

The shuttle peptides used in this study were purchased from GL Biochem (Shanghai, China) with a purity higher than 95%. Peptides were resuspended at a stock concentration of 250 μM with PBS (NaCl 137 mM, Na$_2$HPO$_4$ 10 mM, KH$_2$PO$_4$ 1.8 mM, KCl 2.7 mM, pH 7.4).

## Delivery of GFP-NLS in the presence or absence of Cas RNP

CFF-16HBEge CFTR R553X cells were obtained from and authenticated by the CF Foundation and confirmed mycoplasma negative during the study. 20,000 CFF-16HBEge CFTR R553X cells[14] were plated in each of the 96-well plate the day before the delivery. The components were mixed in PBS with the final concentration of 10 μM recombinant GFP-NLS, 10 μM shuttle peptides with or without 2.5 μM Cas9/ABE8e-Cas9 and 2 μM gRNA. The delivery solution was applied to cells for 60 s before removal and washing of cells. The cells were trypsinized 1–4 h post-delivery and the frequency of GFP positive cells measured using flow cytometry. Approximately 3000 events were recorded by flow cytometry for each sample. Cells were first gated on single cells (FSC-H/FSC-A) followed by gating to remove cell debris (SSC-A/FSC-A). The remaining cells were separated based on GFP level using the FITC-A channel (See Supplementary Fig. 8).

## Delivery of DRI-NLS-Cy5 in the presence of CF sputum

Sputum samples from anonymous CF patients were obtained from the Primary Airway Cell Biobank of McGill University, in Montréal, Canada. Sputum collection followed strict ethical guidelines and patient anonymity was preserved. The study was approved by the IRB of McGill University (IRB # A08-M70-14B). The day before delivery, 20,000 HeLa cells were plated in each well of a 96-well plate. The different components were mixed in RPMI media to a final concentration of 10 μM of DRI-NLS-Cy5 and the indicated concentration of shuttle peptide with or without 10% CF sputum. Where specified, a protease inhibitor cocktail (Millipore, Cat# 535140) at 1:25–1:50 dilution and 2–4 mM PMSF were added to the delivery solution. The delivery solution was applied to the cells. After 5 min the solution was removed and the cells washed. The cells were trypsinized 1–4 h post-delivery and the frequency of Cy5 positive cells measured by flow cytometry. A gating strategy similar to that used for delivery of GFP-NLS (see Supplementary Fig. 8) except that final cells were separated based on Cy5 level using the APC channel.

## Primary cultures of human and rhesus monkey airway epithelia

Human CF and non-CF airway epithelia were provided by the University of Iowa's In Vitro Models and Cell Culture Core. Primary rhesus monkey airway epithelial cells were isolated from post-mortem tracheal tissues (provided by the Tulane National Primate Research Center tissue specimen program). Cultures were prepared in the University of Iowa's In Vitro Models and Cell Culture Core. Cells isolated from trachea or bronchi were grown at the air-liquid interface on collagen coated Costar Transwell polycarbonate filters (catalog # CLS3413, 0.3 μm$^2$ surface area) as reported previously[52]. Cultures were maintained in media supplemented with Ultroser G (USG) or PneumaCult-ALI (StemCell Technologies, Cambridge, MA) and the following antibiotics: penicillin (50 units/ml), streptomycin (50 μg/ml). The cultured cells were maintained at 37 °C in 5% CO$_2$. All primary epithelial cells were well-differentiated (>4 weeks). The study was approved by the Institutional Review Board at the University of Iowa.

## Cas9 and ABE8e-Cas9 RNP formulation and application to epithelia

Shuttle peptides were screened on human airway epithelial cells as previously described[13] (Supplemental Table 1). Shuttle peptides S10 and S315 were tested for their delivery efficiency of ABE8e-Cas9 RNP in rhesus airway epithelia cultured at the air liquid interface. The guide RNA (gRNA) was prepared by combining the crRNA (IDT) and tracrRNA (IDT, Coralville, IA, catalog #1072532) (Supplemental Table 4) at equimolar concentrations (100 μM), annealing at 95 °C for 5 min and renaturation at room temperature as described previously[13]. The RNP was prepared by combining the gRNA and recombinant ABE8e-Cas9 protein in DPBS, and incubating at room temperature (RT) for 15–20 min. The final concentration of gRNA was 2 μM and ABE8e-Cas9 RNP was 2.5 μM. Differentiated ALI cultures were washed 2x with DPBS, RNP mix was applied apically in 50 μl total volume for 3 h, cultures washed again 2x with DPBS, and incubated at 37 °C and 5% CO$_2$. Treatment was repeated three times on day 1, 3, and 6 and then cells were harvested 2 days after the last application. The cells were trypsinized, genomic DNA was isolated with QuickExtract (catalog #QE09050, Lucigen, Middleton, WI) and subjected to HTS to examine the A to G editing efficiency at the *CCR5* locus.

Human airway epithelial cells with a compound heterozygous mutation R553X/L671X were expanded and seeded on collagen-coated Costar Transwell polycarbonate filters and differentiated for more than 3 weeks in PneumaCult-ALI media. The single guide RNA (sgRNA) was ordered from Synthego (Redwood City, CA) and contained 2'O-Methyl chemical modifications at 3 first and last bases and 3'phosohorothioate bonds between the first 3 and last 2 bases. The RNP was prepared by combining a sgRNA and ABE8e-Cas9 protein in DPBS, and incubating at RT for 15–20 min. The final concentration of sgRNA was 2 μM and ABE8e-Cas9 was 2.5 μM. Prior to apical application of RNP with shuttle peptides, the cells were incubated with DPBS for 20 min at 37 °C and 5% CO$_2$. Differentiated ALI cultures were treated as described above on day 1, 3, and 6. On day eight, the short circuit current was measured in Ussing chambers. Then the cell genomic DNA was isolated and subjected to HTS to examine the A to G editing efficiency at the *R553X* locus.

## Electrophysiology

Well-differentiated airway epithelial cultures were mounted in the Ussing chambers and assessed for changes in short circuit current in response to stimuli as previously reported[53]. Airway epithelial cell cultures were pre-stimulated overnight with forskolin (Cayman Chemical, Ann Arbor, MI) (10 μM) and 3-isobutyl-1-methylxanthine (IBMX) (Sigma Aldrich, St. Louis, MO) (100 μM) (F&I). Then cells were mounted in the Ussing chambers, bathed in symmetrical Ringer's solution (135 mM NaCl, 5 mM HEPES, 0.6 mM KH$_2$PO$_4$, 2.4 mM K$_2$HPO$_4$, 1.2 mM MgCl$_2$, 1.2 mM CaCl$_2$, 5 mM Dextrose), transepithelial voltage (Vt) was maintained at 0, and baseline currents were established. CFTR current was measured using the following protocol: amiloride (both Sigma Aldrich, St. Louis, MO) (100 μM), 4,4'-Dilsothiocyano-2,2'-stilbenedifulonic acid (DIDS) (Sigma Aldrich) (100 μM), F&I, followed by the CFTR channel inhibitor GlyH-101 (GlyH).

## Rhesus monkeys

All procedures conformed to the requirements of the Animal Welfare Act, and protocols were approved prior to implementation by the Institutional Animal Care and Use Committee (IACUC) at the University of California, Davis. Eight ~4–5-month-old rhesus monkeys (~1 kg; males and females) were screened then assigned to groups as summarized in Supplemental Table 2. All animals were confirmed seronegative for SpCas9 antibodies immediately prior to study assignment. Group 1 (*n* = 4) evaluated the biodistribution of the intratracheally aerosolized fluorescent protein cargo DRI-NLS-Cy5 with shuttle peptide S10 in the tracheobronchial tree and the lungs, and to identify the airway epithelial cell subtypes incorporating the fluorophore. Two of the four animals receiving the fluorophore alone and served as sham controls. Group 2 (*n* = 4) was designed to examine the editing of the *CCR5* locus after intratracheal aerosolization of the base-editor (ABE8e-Cas9 RNP) and two different shuttle peptides, S10 (*n* = 1) and S315 (*n* = 2). One animal received the base editor alone.

## Labeled protein and ABE8e-Cas9 RNP formulation

All materials were thawed on ice, then DPBS, a shuttle peptide, and DRI-NLS-Cy5 were added in this order and mixed gently by pipetting under sterile conditions. Shuttle peptide was omitted and substituted with DPBS for the control animal. For the gene editor instillation solution, the gRNA duplex was prepared by adding crRNA (CCR5), tracrRNA and Duplex Buffer, and incubated at 95 °C for 10 min. RNP mix was prepared by adding gRNA duplex at room temperature, DPBS, and ABE8e-Cas9, in this order, and incubating at room temperature for a minimum of 15 min, all under sterile conditions. The instillation solutions were administered at room temperature within ~ 30 min of preparation under aseptic conditions.

## Rhesus procedures

Animals were sedated with Telazol IM (5–8 mg/kg) and blood samples collected (~4 ml; hematology, clinical chemistry panel, serum, plasma). Individual animals were placed on the scan bed and a CT scan obtained immediately prior to administration. Instillation of the solution into the airways was performed using the MADgic laryngo-tracheal mucosal atomizer (Fisher Scientific, Catalog No. NC0924493, Teleflex LLC MADgic Laryngo-Tracheal Atomization Device MAD700). The atomizer was carefully inserted into the trachea using a laryngoscope then a prefilled syringe containing 1 ml of the instillation solution was connected to the proximal end of the atomizer and the content was expelled to the airways by applying a pressure to the syringe pestle. This was immediately followed by 400 μl of ambient air to complete the instillation of material left in the atomizer, which was then carefully removed. Animals were repositioned on the scan bed and a post-administration CT scan obtained. All animals were monitored closely during the immediate post-administration period (e.g., respiratory rate and stridor; SpO₂ via pulse oximeter). For two animals in the fluorescent protein group tissues were collected 1–2 h post-administration whereas tissue harvests were performed 7 days after the instillation in the gene editor group. These animals were monitored daily until endpoint. Serum was used at the pre- and post-administration time points to assess inflammatory cytokines using the BioLegend LEGENDplex™ NHP Inflammation Panel (13-plex) multiplex bead-based assay, using fluorescence-encoded beads for flow cytometry. Analysis included IL-6, IL-10, CXCL10 (IP-10), IL-1β, IL-12p40, IL-17A, IFN-β, IL-23, TNF-α, IFN-γ, GM-CSF, CXCL8 (IL-8), and CCL2 (MCP-1).

## Rhesus tissue collection and processing

Following euthanasia (overdose of pentobarbital), the chest cavity was opened, the large vessels were cut at the level of the diaphragm, and the lung vasculature was perfused with PBS with 10 μ/ml Heparin via the right ventricle. The animals instilled with the fluorescent cargo were processed as follows. The lung lobes of animal #1 and #2 (instilled with the higher concentration of the fluorescent cargo) were not flushed with PBS, while the lung lobes of #3 and #4 (lower concentration of the fluorescent cargo) were via tubing fitted in the trachea once with PBS by gravity perfusion to remove any remaining instillation solution. The pluck (lung lobes and heart) was carefully removed from the chest cavity, the left caudal lobe was dissected free and tied off with a suture at the main lobar bronchus, cut below the suture, and fresh tissue as well as ~½ inch of trachea was placed in triple-sealed bags. The remaining trachea was fitted with PE tubing with a three-way stop valve connected to a 50 ml syringe without a plunger prefilled with 4% PF, tied off to create a seal, and the lungs were inflated with the fixative under the gravity of a 30 cm water column. After 10 min, the tubing was removed while the trachea was tied off, and the lungs were placed in triple-sealed bags containing 4% PF. All sealed bags (fresh and fixed tissues) were placed in specialized shipping containers on cold packs for overnight shipment and delivery the next morning.

Upon receipt, the fresh left caudal lobe and proximal part of the trachea were used to collect the airway epithelia brushings. A sterile cytological brush (Ref 25-2199, Puritan Medical Products Company LLC, Guilford, ME; or ConMed Disposable Bronchial Cytology Brush, ConMed Corporation, Utica, NY) was inserted into the main lobar airway or the trachea and brushing was performed by gently rotating the brush several times. Brushings were then collected into 5 ml of PBS in a 15 ml conical tubes by mechanically agitating and swirling the brush to wash off the epithelia. The brushings were centrifuged at 200 g and the pellets were reconstituted in 90 μl of PBS. 40 microliters of the pellet suspension were used for DNA extraction by QuickExtract, 40 μl for flow cytometry or cryo-banking, and 10 μl for the cytospin preparation, DiffQuick stain, and light microscopy. The fixed lungs were placed into 1 liter of 4% PF for additional 24 h, and then processed through the sucrose gradient with 15% and 30% sucrose in PBS for a minimum 48 h each. The lobes were serially dissected into smaller pieces, and alternate sections were embedded in OCT for cryo-sectioning and immunostaining.

The tissues that received the base editor were collected and shipped in a similar manner. For these specimens the left caudal lobe dissected with a large airway and tissue containing parenchyma were placed in Trizol (Invitrogen). The left and right mainstem bronchus was dissected from the trachea at the bifurcation and each lung lobe was separated at the main lobar airway. Trachea, each mainstem bronchus, and each main lobar bronchus was brushed with an individual sterile brush depending on the size of the airway. The left caudal lobe was cut perpendicular to the main axis at approximately mid lobe, and the major airway of the distal part of the lobe was brushed. Brushings were collected into 5 ml PBS in 15 ml conical tubes placed on ice and centrifuged at 200 g for 5 min. The cell pellets were reconstituted in 500 μl of PBS, and the cell suspensions divided into 2 aliquots—one for DNA extraction and one for cryopreservation. The tissues containing large airway and small airways were dissected, placed in RNALater or Trizol and cryopreserved. The selected remaining lung tissue was preserved in 4% PF.

## Immunostaining and quantitative analysis of Cy5+ cells

Cryosections (10 μm thick) adhered to Cryojane adhesive coated slides (catalog # 3P 39475208 CFSA 1X Leica Microsystems) were rehydrated in PBS, permeabilized with 0.1% Triton X-100, and blocked with Superblock (catalog # 37515, Thermo Scientific) and 10% animal sera of a species in which the secondary antibody was raised. Primary antibodies diluted in the staining buffer (PBS, 1% BSA, 0.1% Triton X-100) at 1:50–1:100 dilution was incubated overnight at 4 °C in a humidified chamber, followed by a repeated washing. Secondary antibodies at 1:500–1:1000 were incubated for 60–120 min at room temperature in a humidified chamber, followed by repeated washing. The coverslips were mounted with Vectashield with DAPI and slides were imaged by an epifluorescence (Keyence, BZ-X800 Series, Keyence Corporation, Itasca, IL) or confocal microscopy (Zeiss LSM710 with the Zen software, both Zeiss Group). Where applicable, between 0.3 and 1 μm z-sections images were collected. Image overlays, composites, and pseudo-coloring on some images were prepared using ImageJ/FIJI (ImageJ, NIH). Minimal adjustments of individual color channels in the merged images was performed to better visualize the subcellular structures or decrease the background fluorescence, and was consistent between the experimental and control samples. Co-localization of Cy5-DRI-NLS with specific cell types of interest was examined under high magnification (40x and 63x objective with oil correction). The following staining antibodies were used: Keratin 5 Polyclonal Antibody to stain basal cells (CK5, BioLegend, catalog # 905501, Clone Poly19055), SFTPC Polyclonal Antibody to stain surfactant protein C-secreting cells (Invitrogen, catalog # PA5-71680), Anti-Club Cell Secretory Protein Antibody to stain secretory cells (SCGB1A1, Millipore, catalog # 07-623, polyclonal), and Monoclonal Anti-Tubulin Acetylated Antibody to stain the ciliated cells (AcTub, Sigma-Aldrich, catalog # T6793, clone 6-11B-1). For quantitative analysis of Cy5

positive airway epithelia, the microscopic slides were imaged using an epifluorescence microscope (Keyence, BZ-X800 Series), and manual enumeration of the Cy5+ and DAPI+ cells in the large airways (diameter > 600 μm) and small airways (diameter <600 μm) exhibiting deposition of the instillation solution was performed using the ImageJ/FIJI program. Percent Cy5+ cells were calculated as number of all Cy5+ cells relative to all DAPI+ nuclei in a given airway between the basal membrane and the luminal airway surface.

## High throughput sequencing and analysis

Genomic DNA was isolated from cultured cells or cells isolated by airway brushings using QuickExtract (catalog # QE09050, Lucigen, Middleton, WI) according to the manufacturer's protocol. The target locus was PCR amplified (KAPA DNA polymerase, Roche, Basel, Switzerland) using primers designed to the rhesus *CCR5* locus and the appropriate Illumina forward and reverse adaptors as described[1] (Supplemental Table 3). All HTS primers were purchased from IDT (IDT, Coralville, IA). Unique Illumina barcoding primer pairs were then added to each sample in a second PCR reaction with PhusionU Hot Start DNA Polymerase (Thermo Fisher Scientific). Cycling conditions were: 95 °C for 3 min; 12 cycles of 95 °C for 10 s, 61 °C for 20 s, and 72 °C for 30 s; then 72 °C for 2 min. PCR products were pooled and gel purified using a 1% agarose gel and a QIAquick Gel Extraction Kit (QIAgen) to remove primer dimers. Purified PCR products were next quantified and sequenced using a single-end read of 200-300 bases on the Illumina MiSeq instrument using the manufacturer's protocols. Following high throughput sequencing (HTS), the sequencing reads were demultiplexed using MiSeq Reporter (Illumina) and aligned to the appropriate reference genome as previously reported[54]. Base substitution and indel frequencies were assessed using the software package CRISPResso2[54], counting indels of ≥1 base occurring in a 20-base window around the ABE nicking site. Indels were defined as detectable if there is a significant difference (Student's two-tailed *t* test, *P* < 0.05) between indel formation in the treated sample and untreated control. For each *CCR5* modification we determined the target base editing frequencies, bystander edits, and indel frequency[1,54]. Note that for results obtained for the human R553X mutation, the airway epithelia were compound heterozygous for this CFTR mutation. Therefore, all cells contained "wild type" sequence (at the genomic locus being assessed) for one allele and had the R553X mutation on the other allele. The allelic editing frequency for this heterozygous target was calculated by the following equation: (% of alleles with the desired edit-50)/50 * 100. This mathematical adjustment provides a meaningful "per cell" estimate of correction. The HTS data are accessible through the NCBI Sequence Read Archive database under Bioproject accession code PRJNA1043615.

## CIRCLE-seq

Circularization for In vitro Reporting of Cleavage Effects by sequencing (CIRCLE-seq) was performed and analyzed as described previously[17]. Genomic DNA was isolated from rhesus monkey lung tissue using the QIAgen Gentra Puregene kit (catalog # 158063) according to manufacturer's instructions for tissue extraction. DNA was sheared, circularized, and cleaved with Cas9 nuclease RNP. To generate the RNP, synthetic sgRNA with the sequence used to target rhesus monkey CCR5 was ordered from Synthego at 1.5 nmol scale using their standard chemical modifications, 2'O-Methyl modifications for the first three and last three bases, and phosphorothioate bonds between the first three and last two bases. Guide RNA was diluted to 9 μM in nuclease-free water and re-folded by incubation at 90 °C for 5 min followed by a slow annealing down to 25 °C at a ramp rate of 0.1 °C/s. The sgRNA was complexed with Cas9 nuclease (NEB; M0386T) via a 10 min room temperature incubation after mixing 5 μL of 10x Cas9 Nuclease Reaction Buffer provided with the nuclease, 4.5 μL of 1 μM Cas9 nuclease (diluted from the 20 μM stock in 1x Cas9 Nuclease Reaction Buffer), and 1.5 μL of 9 μM annealed sgRNA. Circular

DNA from rhesus monkey lung tissue was added to a total mass of 125 ng and diluted to a final volume of 50 μL. Following 1 h of incubation at 37 °C, Proteinase K (NEB; catalog # P8107S) was diluted 4-fold in water and 5 μL of the diluted mixture was added to the cleavage reaction and incubated for 15 min at 37 °C. RNP preparation and DNA cleavage was performed as described in[55]. DNA was A-tailed, adapter ligated, and USER-treated, and PCR-amplified as described in[17]. Following PCR, samples were loaded on a preparative 1% agarose gel and DNA was extracted between the 300 bp and 1 kb range to eliminate primer dimers before sequencing on an Illumina MiSeq as 50% of the run, with an uncleaved negative control sample from rhesus monkey DNA making up the remaining 50%. Paired end reads of 150/150 were sequenced. All primers were purchased from IDT (IDT, Coralville, IA). Data was processed using the CIRCLE-seq analysis pipeline located at https://github.com/tsailabSJ/circleseq[8], and aligned to the macaque genome with parameters: "read_threshold: 4; window_size: 3; mapq_threshold: 50; start_threshold: 1; gap_threshold: 3; mismatch_threshold: 6; merged_analysis: True".

## MAD7 nuclease delivery to Ai9 mice

In vivo delivery to mice was performed at TransBIOTech according to the institutional guidelines and as described previously[13]. The mouse study was approved by the CEGEP de Lévis Animal Care Committee (Lévis, Quebec, CA; Approval 024-20) and complied with CACC standards and regulations governing the use of animals for research. Following arrival, Ai9 B6.Cg-Gt(ROSA)26Sor^tm9(CAG-tdTomato)Hze/J mice (from Jackson Labs, Strain #007909; on a genetic background Strain #000664 C57BL/6 J) were subjected to an acclimation period of 7 days before beginning the study. Mice were maintained in the animal rooms on a regular dark/light cycle, at a controlled ambient temperature and humidity. Tap water and standard certified commercial rodent diet (Envigo 2018) was provided *ad libitum* except during designated procedures requiring the handling of animals outside of their housing cages (such as dosing). Briefly, shuttle peptide and MAD7 RNP were formulated in PBS and delivered to 8–10-week-old, male and female Ai9 mice[19]. A volume of 50 μl shuttle peptide (S10, 40 μM) was formulated with nuclease and gRNA in the RNP preparation of 1.33 μM MAD7 and 2 μM gRNA. The MAD7 gRNA sequence used was: GTATAATGTATGCTATACGAA; PAM: CTTC. Mice were anesthetized with ketamine/xylazine intramuscularly (87.5 mg/kg ketamine/12.5 mg/kg xylazine) and the shuttle peptide and MAD7 RNP administered intranasally. Mice were euthanized at indicated time points. Tissues were collected and fixed in cold paraformaldehyde (4% in PBS) for 4 h and then incubated overnight in 30% sucrose (in PBS) at 4 °C. Lungs were embedded in OCT compound in a dry ice-cold bath of isopentane.

## Microscopy and quantitative analysis of mTomato+ cells

Cryosections (5 μm thick) were rehydrated in PBS, mounted with Prolong Glass NucBlue (Invitrogen), and imaged within 1–7 days using an automated slide scanner (PANNORAMIC MIDI II, 3DHistech Ltd.). Quantification of mTomato fluorescence in digital images was performed using the CellQuant module from CaseViewer software from 3DHistech. Airway epithelia between the basal membrane and the luminal airway surface were selected as regions of interest within which the number of mTomato + nuclei were counted against all DAPI + nuclei and expressed as a percent.

## Mice pulmonary toxicity study

Studies in mice were performed in accordance with the University of Iowa's Institutional Animal Care and Use Committee (IACUC, Animal Protocol # 7072030-008) and in accordance with National Institutes of Health guidelines. 6–8 week old male and female C57BL6/J mice (Jackson Labs, Strain # 000664) were acclimated for 7 days. Mice were maintained in the Thoren caging units in animal rooms on a regular dark/light cycle, at a controlled ambient temperature and humidity.

Mice received two 50 µl intranasal instillations on 2 consecutive days under light isoflurane anesthesia (one 50 µl dose per day). The instillation solution consisted of one of the following: DPBS (control), RNP only (RNP), S10 (peptide S10 at 40 µM final concentration in DPBS), S315 (peptide S315 at 40 µM final concentration in DPBS), S10 + RNP (peptide S10 at 40 µM and RNP at 2.5/2 µM final concentration in DPBS), S315 + RNP (peptide S315 at 40 µM and RNP at 2.5/2 µM final concentration in DPBS); the naïve control group received no treatment. Groups of 8 mice per group (naïve mice $n = 4$) were euthanized and examined on day 1 and 7 after the second instillation. Total and differential cell counts were determined from the BAL fluid that was collected by lavaging the lungs with 2 ml DPBS at 25 cm water column pressure. Cytospins prepared from BAL were stained with a DiffQuick (Siemens #10736131) and 200 cells per slide were counted to determine the percentual proportion of different cell types. For analysis of pro-inflammatory cytokines and chemokines, the lung tissues were homogenized in Trizol, the isolated RNA (Zymo Research, Irvine, CA) was transcribed to cDNA (AppliedBiosystems, ThermoFisher Scientific), and the mRNA levels determined utilizing RT qPCR primers and methods as reported previously[13]. For lung histopathology, the fixed tissues were paraffin-embedded and ~4 µm thick sections stained with hematoxylin and eosin (H&E). Tissues were examined by a board certified veterinary pathologist using the post-examination method of masking group assignment[56]. Tissues were scored for the presence of cellular inflammation (excluding iBALT), epithelial hyperplasia, and necrosis: 0 – none; 1 – isolated to multifocal, <25%; 2 – multifocal to coalescing, 26%–50%; 3 – coalescing to lobar, 51–75%; and 4 – multilobar, >75% (fields of 200x magnification).

## Statistics and reproducibility

The number of the animals per treatment group are listed in the respective experimental sections in the Methods and/or in the figure legends. The number of mice per group was determined empirically to allow detection of differences between two groups and ranged between 2 and 8 mice per group. The number of rhesus monkeys was between 1 and 2 per group and was guided by the ethical decision making in conjunction with respective guidelines. Males and female rhesus monkeys and mice were randomly assigned to treatment groups. No animals or data were excluded from analysis. The investigators were not blinded to allocation during experiments and outcome assessment, unless stated otherwise in the respective experimental section. Student's two-tailed $t$ test, Kruskal–Wallis test by ranks, or one-way analysis of variance (ANOVA) with Tukey's multiple comparison test were used to analyze differences in mean values between groups. Results are expressed as mean ± SEM. $P$ values ≤ 0.05 were considered significant.

## Reporting summary

Further information on research design is available in the Nature Portfolio Reporting Summary linked to this article.

## Data availability

All relevant data supporting the key findings of this study and any associated accession codes and references are available within the article and in the Supplementary Information files or from the corresponding authors upon request. Source data for all figures are provided with the paper. High-throughput sequencing data have been deposited in the NCBI Sequence Read Archive database under Bioproject accession code PRJNA1043615. Source data are provided with this paper.

## Code availability

The code used for analysis of HTS data is available at https://github.com/pinellolab/CRISPResso2. The code used for the CIRCLE-seq analysis is located at https://github.com/tsailabSJ/circleseq.

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

## Acknowledgements

These studies were supported by a Collaborative Opportunity Fund project provided through the NIH Somatic Cell Genome Editing (SCGE) Consortium. This was a collaborative study between delivery and editor projects (UG3 HL147366 (P.B.M.); and U01 AI142756 (D.R.L.)) and the Nonhuman Primate Testing Center for Evaluation of Somatic Cell Genome Editing Tools (U42 OD027094 (A.F.T.)). Information about these studies is also provided in the SCGE Toolkit, supported through the SCGE Dissemination and Coordinating Center (https://commonfund.nih.gov/editing), which is a platform housing data generated across all consortium initiatives[57]. Studies were also supported by U24 HG010423 (P.B.M.), P01 HL152960 (P.B.M.), the Center for Gene Therapy of Cystic Fibrosis: P30 DK-54759 (P.B.M.), the Cystic Fibrosis Foundation (MCCRAY15XX0 (P.B.M.), FELDAN21W0-SC (D.G.)), and the base operating grant for the California National Primate Research Center (P51 OD011107 (A.F.T.)). D.R.L. is supported by NIH P01 HL152960, U24 HG010423, UG3 AI150551, R35GM118062, RM1HG009490, and HHMI. We acknowledge the support of the University of Iowa In Vitro Models and Cell Culture Core and the Comparative Pathology Core. G.A.N. was supported by the Helen Hay Whitney postdoctoral fellowship, K99 award HL163805, and CFF PTAC Pioneer award 23XX0. F.C. is supported by CQDM Couture-QL-274 and a member of the Institute of Nutrition and Functional Foods of University Laval (Québec, Canada) and of the Research Center of the CISSS de Chaudière-Appalaches (Lévis, Canada). P.B.M. is supported by the Roy J. Carver Charitable Trust. In vivo imaging (CT scans) was performed with instrumentation funded by the NIH (S10 grant #RR025063 (A.F.T.)). We thank Teresa Ruggle of the University of Iowa Design Center for assistance with graphic design, as

well as Ms. Isabelle Bolduc from TransBIOTech for technical assistance. The authors thank the animal care and veterinary staff at the Primate Center (particularly Dr. Kari Christe); Dr. Peter S. Thorne and Dr. Nervana Metwali of The Human Toxicology and Exposomics Laboratory, the University of Iowa (Iowa City, IA), for the endotoxin assay and analysis; Robert Blair and colleagues at the Tulane National Primate Research Center for providing post-mortem rhesus airway tissues; and Dr. Patrick Ten Eyck for statistics support. This article is subject to HHMI's Open Access to Publications policy. HHMI lab heads have previously granted a nonexclusive CC BY 4.0 license to the public and a sublicensable license to HHMI in their research articles. Pursuant to those licenses, the author-accepted manuscript of this article can be made freely available under a CC BY 4.0 license immediately upon publication.

## Author contributions

P.B.M., A.F.T., G.A.N., D.R.L., K.K., D.J.H.-O., D.K.M., S.H., X.C. and D.G. conceived and designed the experiments. A.F.T. performed all rhesus monkey studies. D.J.H.-O., K.K., X.C., S.T., G.A.N., S.H., G.R.R., B.T., G.K., B.P.Y., D.K.M., S.K., F.C., H.B.F., M.H., T.J.G., J.R. and C.W.-L. carried out experiments and analyzed data. F.C. performed the Ai9 mouse study. P.B.M., A.F.T., G.A.N., X.C., K.K., S.T. and D.G. wrote the manuscript.

## Competing interests

P.B.M. is on the SAB and performs sponsored research for Spirovant Sciences, Inc. D.G. holds equity in Feldan Therapeutics. X.C., M.H. and D.G. are employees of Feldan Therapeutics. S.H., H.B.F. and J.R. were employees of Feldan Therapeutics. D.G. is co-inventor on patents and patent applications filed by Feldan Bio Inc. on the Shuttle peptide technology. D.R.L. is a consultant and equity holder of Beam Therapeutics, Prime Medicine, Pairwise Plants, Chroma Medicine, and Nvelop Therapeutics, companies that use or deliver gene editing or epigenome modulating agents. D.R.L. and G.A.N. are co-inventors on patent applications filed by the Broad Institute on base editing and its applications. The remaining authors declare no competing interests.
