## [Peer Review File · Nature Communications]

reviewers' Comments:

Reviewer #1:

Remarks to the Author:

Katarina Kulhankova and colleagues reports a peptide S315, derived from a formerly reported S10 which allows to deliver Cas9 base editor RNP in primary airway epithelia cells.

The base editing was tested in vitro, both in human and rhesus airway epithelial cells in vivo in rhesus and mice. The authors measure the efficiency of delivery of both the original S10 peptide and of the newly developed S315. Different readouts are employed ranging from base editing efficacy in genomic loci (B2M, CCR5 and CFTR) or through fluorescence of Cy5 dye, or indel formations through fluorescence detection.

The study is of potential interest in the field of genome editing where tools of delivery are highly needed, nonetheless the study lack of homogeneity mainly due to a variety of cargo used (Cas9 base editor, Cy5 dye and Cas12 nuclease) using relative different readouts. The focus is also lost by experiments done with S10 and S315 not always comparatively analyzed rather tested individually.

Major points:

1) In figure 2 the base editing efficiency was analyzed in 2 different loci in human and rhesus airway cells. The study can be highly improved with the analysis of additional loci including benchmarks used in the base editing studies (e.g. HEK sites, TRAC and more). In the same figure the editing efficacy for each locus is reported as a general % of base editing not reporting the % of modification of nucleotides located in the window of ABE8e activity (similar to the structure of analysis in fig. 7).

2) Given the toxicity frequently associated with peptide carriers the analysis should be performed for S315. In a former study (Nat Comm 2019) the same group performed S10 toxicity analysis in lungs obtaining encouraging results.

3) A comparative analysis with other delivery tools highly used with base editor (RNA through LNP) at least in in vitro cultures would be a fundamental information to the readers. Beyond the efficiency the comparative analysis with other tools could provide information on the bystander edits. Bystander activities of base editors often limits their broader use in particular in exons where unwanted modifications are difficult to control. The delivery of ABE through S315 results in the same bystander edits as other delivery tools (RNA or endogenous expression through DNA vectors)?

Minor points:

1) Figure 3 (Cy5 with S10) can be moved to the suppl section. This may improve the focus of the study which is a bit diluted throughout the manuscript with the analysis of diverse cargos (Cy5, ABE and Cas 12a nuclease) with S10 or S315.

2) Even though data in Figure 6 using Cas12 nuclease with S10 provide an interesting indication on the persistence of edited cells in airways, they are not addressing the residency of base edited cells through S315. Lung tissues treated with reagent with different properties (S10-nuclease vs S315 base editor) may have very different destiny. Suggest moving to suppl.

3) why the authors do not provide data on the potential bystander edit (A11) in figure 7? The position of the A close to the target site is highly indicative of a potential bystander site. The authors very likely have already the results in their sequence analysis.

4) In figure 7 the % of editing referred to a single allele (value of the editing multiplied by 2) should be clearly explained in the text to avoid confusion

5) in figure 7 the editing efficacy does not correlate with CFTR recovery. The authors should discuss this discrepancy

6) figure 4: why a single inhalation was performed?

7) Discussion is needed to elaborate on the discrepancy between efficiency of Cy5 and ABE delivery in vivo in rhesus monkey

Reviewer #2:

This is a new manuscript submission by Kulhankova et al. entitled, "Shuttle Peptide Delivers base editor RNPs to Rhesus Monkey airway epithelial cells in vivo". The work builds on a previous publication identifying shuttle peptides specially the amphiphilic S10 when combined with Cas-RNP can enhance gene editing in human and mouse airway cells. Here, the authors optimized their previously reported S10 shuttle to derive the S315 peptide and used it for adenine base editor RNP (ABE8e-Cas9 RNP) delivery. Using a peptide screen in well differentiated human airway cultures, they identified several new peptides that delivered Cas9 RNP more efficiently than S10, with specific focus on the most highly efficient one S315. Using a rhesus monkey model, intratracheal aerosol delivery of the S10 peptide cargo (Cy5-labelled) resulted in Cy5 fluorescence (peptide distribution) throughout the airways and as deep into the lungs as the alveoli. With this promising result, they then targeted a safe harbor site CCR5 using S315 and show editing efficiency on average ~5% in the airways of these monkeys, with no signs of toxicity or proinflammatory responses. Persistent (up to 12 months) expression of gene editing in the airways was observed using a transgenic mouse model (Ai9 ROSA26 tdTomato), however this was determined using the original S10 shuttle peptide. And finally, codelivery of the S315 and ABE8e-Cas9 RNP in primary human airway epithelial cells heterozygous for the mutation R553X showed restoration of CFTR function and achieved higher levels of editing than the S10 peptide.

Overall, this study builds on their previous work on S10 published in 2019, that now compares S315 with S10 and their ability to improve base editing in relevant animal and human models. There is enthusiasm for this work however, there are some important concerns/clarifications that needs to be addressed.

1. How stable is the S315 or any of the shuttle peptides in a pro-inflammatory environment? The study shows great efficiency at targeting cells along the respiratory tree but in the context of disease such as CF, the airways is a highly inflammatory milieu and therefore understanding the stability and targeting efficacy in this context is important especially since the last figure suggests a potential application of this system for CF. One possibility is to test stability in CF patient sputum which is enriched with proteases.
2. The rationale for using S10 in the biodistribution studies and not S315 is unclear. The argument is that S315 is a better shuttle peptide as the efficiency of targeting is much higher and therefore, for both biodistribution and toxicity studies, why not evaluate with S315 (irrespective of GFP delivery activity)?
3. The authors do note in the discussion that the rhesus monkey experiments only tests 1 bolus (not serial) and 1 concentration (dose) to determine editing efficiency. Thus, the sample size is very small. For biodistribution, using this model is quite relevant to human systems. However, biodistribution here is in the context of a "normal" lung (no inflammation/injury). It would be interesting to see if this is also true in an injured setting, and as in the mouse study, whether "persistence" is altered in a setting where there would more cell turnover (ie. injury). Second, it is not clear how the dose was determined and more importantly, how serial dosing or higher concentrations may affect efficacy (albeit improve targeting efficiency) but also toxicity and pro-inflammatory responses which would be interesting in the primate model. Alternatively, for concerns with costs associated with the primate model, one possibility it to use the mouse model, perhaps even the humanized CF mouse model?.
4. The recombination for tdTomato expression appears to be exclusive to the airways (no alveoli) in Fig6, whereas in Fig3, editing was observed in the alveolar regions. Can the authors explain?

5. In the human CF airway epithelial cell model (Fig 7), while editing efficiency was higher with S315 than S10 (almost 2-fold), functional correction is the same. Can the authors explain why? Wouldn't S10 be a better candidate peptide for CF R553X base editing in this case? Is the goal here to improve editing efficiency (# of cells targeted) or functional outcome from editing? If the later, S10 seems to be just as good at delivering base editors in this case. It was not clear how many patient cell lines were used here? There are many primary cell biobanks to test additional lines and potentially other mutations.

6. It is unclear if there were off-target effects with improved editing efficiency. Please clarify.

Reviewer #3:

Remarks to the Author:

The manuscript by Kulhankova and colleagues reports an array of experiments validating a pre-clinical strategy – pairing a base editor enzyme with a novel delivery approach – that has translational potential for therapeutic correction of cystic fibrosis (CF). Although small molecule drugs can currently benefit CF patients with certain mutations, there are still many who harbor mutations that underlie a form of CF that cannot be addressed with existing drugs. One such mutation is R553X, which is the focus of the present work. Prior work by the Liu/McCray team has demonstrated that an adenine base editor (ABE) can enact a genome-level correction for this mutation that would be anticipated to provide clinical benefit in any tissue receiving such a correction. However, delivery of genome editing enzymes to the lung epithelium has not yet been attained in the clinic. Although some pre-clinical work suggests lipid nanoparticles (bearing mRNA or RNP cargo) may hold promise for therapeutic editing of the lung epithelium, delivery to the lung is by no means a solved problem. To address this unmet need, Kulhankova and colleagues have applied (and improved upon, via novel peptide discovery) a strategy previously established by the Guay/McCray team: mixing pre-formed ABE enzymes with amphiphilic peptides that facilitate productive intracellular delivery of RNP enzyme cargo.

In the present manuscript, these two key technologies (therapeutic base editing and peptide-mediated enzyme delivery) are shown to work well together in multiple contexts: human and primate tissue culture models of airway epithelium, the same model using CF-patient derived cells, as well as in vivo administration for editing of the lung epithelium in primates. Additional evidence for the translational approach is provided by complementary experiments: assessment of the cell types being reached using their delivery approach as well as an assay evaluating the persistence of edited cells over the course of a year in a fluorescent reporter mouse model. The study concludes with a thoughtful and precise analysis of the likely clinical endpoints pertinent for development a functional CF therapy. Encouragingly, this analysis accurately finds that the present pre-clinical approach holds substantial promise for successful clinical translation. The authors would be well justified to express even more optimism regarding their capacity to advance the field of genetic therapy for CF: a virally-vectored traditional gene therapy approach has apparently languished for years, indicating the need for new and innovative approaches to address the patients who cannot benefit from current CF drugs.

I recommend publication of this manuscript after a two minor points are suitably addressed. As is described in detail below, I have doubts regarding certainty of the conclusions associated with the GFP/Cas9 “competition” assay reported in Fig. 1c. Also described in detail below is a concern regarding the processing/reporting of sequencing data is discussed. With these two issues having been corrected or clarified, I believe this manuscript will be an outstanding candidate for publication by Nature Communications. I have also provided feedback that strives to strengthen the manuscript, but these additional points may be prioritized according to the authors’ and editor’s discretion.

Below, main-text line numbers will be marked as M###; supplementary information line numbers will be marked as S###

Major concerns

1. M109 / Fig. 1c / Supp. Fig. 1 (experiment & conclusions)

I am not sure these experimental results can only be explained via the proposed mechanism. The authors suggest that diminished GFP delivery (by peptide S10) is due to the peptide’s

"susceptibility to inhibition by Cas9 RNP", and this finding is extended (if I understand correctly) to insinuate that formulation containing S10 and Cas9 would somehow be harmful for delivery of Cas9. But previous work has clearly shown that S10 is perfectly capable of facilitating Cas9 delivery both in tissue culture models and in vivo. The assay reported in Fig. 1c is most directly assessing each peptide's ability to deliver GFP, and how that ability is impacted by the presence of Cas9 RNP. I don't think these results can rule out a model wherein S10 peptide favors non-covalent association with Cas9 RNP (as opposed to GFP). In contrast, peptide S315 could favor interactions with GFP (along with, or instead of with Cas9 RNP). This is an alternative mechanism/model that explains the results shown in Fig. 1c, and it doesn't have particular explanatory value for the manuscript writ large. I suggest the authors either remove their narrow mechanistic interpretation of the experiment, augment their explanation to include the alternative model, or perform an additional experiment to narrow the possibilities. Regarding the latter, one such experiment could be a Cas9 RNP-mediated genome editing assay that is performed using either peptide, in the absence or presence of GFP. If my proposed model is correct, editing levels will not be impacted by the presence of GFP when S10 is used, while S315-mediated editing will be constant or diminished in the presence of GFP. To be clear, I don't think these experiments – that of Fig. 1c and/or my proposed experiment – are essential for the manuscript's overall conclusions. Mechanistic understanding of delivery is valuable but not necessary.

2. S249-250 (and legend of Fig. 7)

The supplement states: "The allelic editing frequency was calculated by the following equation: (% editing- 250 50)/50." Make sure this equation is reported consistently: the supplement matches text in Fig. 7, but the Fig. 7 legend diverges in that it has "* 100" appended. I don't think this is likely to cause substantial confusion, but it's nice to be consistent. In contrast, I think the typical reader may actually be confused about how/why 52.5% editing represents ~5% editing. It would be helpful to add a brief clarifying statement in the Results and/or Methods sections noting that all cells contain "wild type" sequence (at the genomic locus being assessed) for one allele, and bear the R553X mutation on the other allele – hence the mathematical adjustment, which provides a meaningful "per cell" estimate of correction.

Suggestions

Discussion: It might be valuable to highlight the potential strengths of this approach as it is moved towards the clinic. In addition to highlighting the strengths of non-viral CRISPR delivery (for decreased immunogenicity and genotoxicity), there may be advantages in terms of GMP & CMC. GMP CRISPR RNP found its way to clinical use extremely rapidly (CRISPR Therapeutics' SCD trial) and peptide should be quite amenable to GMP manufacture. This is a major real-world advantage in contrast to viral manufacture, which can be quite complicated and prone to complications & delays.

M63 Regarding lipid-based delivery of RNP to the lung epithelium, perhaps Cheng et al. 2020 (<https://www.ncbi.nlm.nih.gov/pmc/articles/PMC7735425/>) should be considered. RNP delivery to the lung is reported in Fig. 4f, and the microscopy might be sufficiently compelling to warrant mention here.

M67 Consider revision to "avenues for Cas nuclease and base editor RNP deliery"; "Cas" is essentially an adjective, and "nuclease" is the appropriate term to contrast against "base editor".

M94 Do you really intend to describe lysine as a "di-basic" residue? I believe it should be described as "basic".

M121 Consider revision to "greater than with S10"

M144 Consider revising to "(Fig. 3b, top)"

M149 Consider revising to "(Fig. 3b, middle)"

M152 Consider revising to "(Fig. 3b, bottom)"

M168 / M188 / S153 It's difficult to assess how much material is being administered because the

relevant values are not reported together in the same place. For example, M168 reports concentrations, but the volume is only reported at S153.

M191 For readers who are not pulmonologists, it would be helpful to include a very brief explanation for the role of ground glass in CT scans. Furthermore, it's probably worth presenting CT in its expanded form (computed tomography) the first time it is introduced.

M215 Consider revision to "There was a statistically significant decline in the number of tdTomato+ cells from 7 days to 12 months, but no such decline was apparent when comparing the 7 day time point to other time points up to 6 months."

M222 Consider citing Krishnamurthy 2021, since it provides helpful context regarding the use of the NG PAM for R553X base editing.

M250-251 Consider revision to "The present demonstration"

M271-272 Consider revision to "While ionocytes express the highest levels of CFTR transcripts, they are a rare cell type and their function is still a subject of study, as is their role in genetic therapies for CF."

M279-280 Consider rephrasing "targeted cell types" – this seems to imply deliberate targeting of particular cell types, which isn't a focus of the approaches employed by this study. Perhaps "preferentially-edited cell types" would be more appropriate.

M287 Consider revision to "The strategy presented here has strengths and limitations." The current wording – "This study has advantages" – seems to reflect the study design, not the pre-clinical therapeutic strategy.

M303 Consider revision to "Our results in Ai9 mice following editing using MAD7 nuclease"

Fig. 1b For the boxes around basic residues, consider a color pair other than red & green, since it's not colorblind friendly. In this case, red & black should suffice. For the arrow noting the transplanted leucine, the arrowhead should be pointing at the red leucine, not the black leucine marked as "conserved".

Fig. 4c Consider adding "CCR5" to this panel, preferably as part of the y-axis.

Fig. 7c As above, consider adding "R553X locus" to y-axis.

S144 Although sterile conditions are noted for various steps, there is no mention of sterile filtration of the CRISPR protein(s). If the recombinant proteins (produced in *E. coli*) are not sterile, it is unclear how much the downstream steps matter.

S147 "Procedures" heading is too broad to be useful; revise to a more specific heading, such as "Rhesus macaque in vivo procedures".

Reviewer #4:

Remarks to the Author:

Comments to authors

This is a very well-written manuscript, the figures are clearly presented and the study represents an advance in the field. Strengths include the use of nonhuman primates to extend from in vitro and mouse studies, which provides the potential for further development. Weaknesses include uncertainty if the 5% transduction and editing efficiency demonstrated with the monkey study will be long-lived, or provide a physiological amelioration of the CF condition. The study would benefit from monkey trials that last beyond 1 week.

Specific comments

Abstract

Line 29: 5.3% is at the very low end of the therapeutically beneficial range cited later in the ms. While persistence was documented in mice, the manuscript would be far stronger if monkey studies were carried out beyond 1 week.

Line 114: it is not clearly explained why the B2M locus was targeted for base editing.

Line 121: a 9% in vitro efficiency is difficult to translate to a much less favorable in vivo environment.

Line 144: it would be helpful to provide additional letter labels to the panels of Figure 3b, it is not immediately obvious which panel of the figure is being referred to since there are multiple components of Fig 3b.

Lines 163-4. There is a very wide range of labeling efficiencies. This was not discussed as a limitation of the study but is it significant.

Lines 213 and 215 seem to convey contradictory messages. 215 says tdTomato was persistent for 12 months but lines 214-5 say that there was a decline in the number.

Lines 234-6. While Fig 7 shows cellular impact, the study would be strengthened by demonstration of a physiological therapeutic impact on animal well-being.

Discussion

Line 262: given that the therapeutic range stated is 5-50% restoration needed, it is important to note that the short-term monkey study was at the low end of this range.

Line 281: for me it wasn't clearly explained why CCR5 was targeted.

Reviewer #1

Katarina Kulhankova and colleagues reports a peptide S315, derived from a formerly reported S10 which allows to deliver Cas9 base editor RNP in primary airway epithelia cells.

The base editing was tested *in vitro*, both in human and rhesus airway epithelial cells *in vivo* in rhesus and mice. The authors measure the efficiency of delivery of both the original S10 peptide and of the newly developed S315. Different readouts are employed ranging from base editing efficacy in genomic loci (B2M, CCR5 and CFTR) or through fluorescence of Cy5 dye, or indel formations through fluorescence detection.

The study is of potential interest in the field of genome editing where tools of delivery are highly needed, nonetheless the study lack of homogeneity mainly due to a variety of cargo used (Cas9 base editor, Cy5 dye and Cas12 nuclease) using relative different readouts. The focus is also lost by experiments done with S10 and S315 not always comparatively analyzed rather tested individually.

Major points:

Q1. In figure 2 the base editing efficiency was analyzed in 2 different loci in human and rhesus airway cells. The study can be highly improved with the analysis of additional loci including benchmarks used in the base editing studies (e.g. HEK sites, TRAC and more).

R1. The goal of Figure 2 is to assess whether S315 enables higher base editing than S10, as observed with nuclease editing at the *CFTR* locus in Fig. 1a. The *B2M* locus was selected because it is a ubiquitously expressed gene product in diverse cell types including airway epithelia and has a single adenine in the base editing window. The *CCR5* locus was selected as a safe harbor for a first *in vivo* editing experiment in the rhesus monkey. This gRNA was specifically designed for this study as no gRNA was available for base editing in rhesus monkey cells. We modified the text to clarify these choices (see responses to queries 43 and 50 below).

Q2. In the same figure the editing efficacy for each locus is reported as a general % of base editing not reporting the % of modification of nucleotides located in the window of ABE8e activity (similar to the structure of analysis in fig. 7).

R2. Thank you for this suggestion. The revised Figure 2 includes the same bystander information (see Fig. 2a, c).

Q3. Given the toxicity frequently associated with peptide carriers the analysis should be performed for S315. In a former study (Nat Comm 2019) the same group performed S10 toxicity analysis in lungs obtaining encouraging results.

R3. In response to this query, we performed additional studies in mice evaluating pulmonary responses to shuttle peptides alone and peptide + ABE RNP in combination. The results of these studies are now described in the results section and in a new Supplementary Figs. 4, 5, 6. See lines 246-256.

Additional cytokine/chemokine assays were performed on rhesus sera pre- and post-treatment as described in the revised manuscript (See main manuscript lines 177-183 and Suppl Information lines 182-183). As noted in the submitted manuscript all complete blood counts (CBCs) and clinical chemistry panels, which represent standard routine monitoring in general, were within normal limits. These are routine clinical parameters monitored and as stated all of the values were within the normative range for animals in this age group. Generally, for studies which always include CBCs and clinical chemistry panels where there is no evidence of any findings they are not included beyond a statement they were within normal limits.

We modified the results to state: “They were monitored daily and showed normal activities and food intake. Complete blood counts (CBCs) and clinical chemistry panels prior to and post-administration were all within normal limits. Circulating inflammatory cytokines were assessed at the pre- and post-administration time points by cytokine bead array including IL-6, IL-10, CXCL10 (IP-10), IL-1 β , IL-12p40, IL-17A, IFN- β , IL-23, TNF- α , IFN- γ , GM-CSF, CXCL8 (IL-8), and CCL2 (MCP-1). No statistically significant differences between the pre- and post-administration blood samples were detected.”, Main manuscript lines 177-183.

In Lines 356-359 we state: “Gene editing was achieved without evidence of toxicity in rhesus monkeys during the study period. Additional pulmonary toxicity studies in mice demonstrated mild cellular inflammatory changes that were resolving by day 7 post instillation.” See Supplementary Figs. 4-6.

Q4. A comparative analysis with other delivery tools highly used with base editor (RNA through LNP) at least in *in vitro* cultures would be a fundamental information to the readers. Beyond the efficiency the comparative analysis with other tools could provide information on the bystander edits. Bystander activities of base editors often limits their broader use in particular in exons where unwanted modifications are difficult to control. The delivery of ABE through S315 results in the same bystander edits as other delivery tools (RNA or endogenous expression through DNA vectors)?

R4. The primary focus of this study was to investigate the utility of the peptide mediated RNP delivery strategy *in vivo* in the rhesus monkey model. We agree that examining the bystander edits is of interest. We note that beyond the delivery method, the activity of the selected base editor can substantially influence the frequency of bystander editing. This has been demonstrated elegantly by Kiran Musunuru and colleagues in base editing studies in the liver.

We revised figures to report bystander edits in more detail for the guide RNAs used in this study (Revised Figs. 2a,c, 6b). We are interested in alternative ABE delivery systems (including viral and non-viral vectors), but to date have not identified an alternative system for such comparisons. Therefore, comparing the delivery efficiency or bystander edits with other non-viral and viral *in vitro* transfection studies has limitations and is beyond the scope of this study.

We also performed an analysis of off target gRNA dependent DNA editing following the *in vivo* delivery of CCR5-targeted ABE8e RNPs to rhesus monkeys using the S315 peptide. We observed no evidence of significant off target editing in the amplified candidate sites and provide these data in new Supplementary Fig. 2 and Supplementary Tables 4-6.

Minor points:

Q5. Figure 3 (Cy5 with S10) can be moved to the suppl section. This may improve the focus of the study which is a bit diluted throughout the manuscript with the analysis of diverse cargos (Cy5, ABE and Cas 12a nuclease) with S10 or S315.

R5. We can move these data to the supplement section but prefer to keep this figure in its current position as it helps provide a visual assessment of the cell types targeted using a peptide mediated delivery approach. While not a focus of the paper, the peptide delivery approach may have applications for the delivery of diverse cargoes, a point that may be of interest to readers.

Q6. Even though data in Figure 6 using Cas12 nuclease with S10 provide an interesting indication on the persistence of edited cells in airways, they are not addressing the residency of base edited cells through S315. Lung tissues treated with reagent with different properties (S10-nuclease vs S315 base editor) may have very different destiny. Suggest moving to suppl.

R6. As suggested, we moved Fig 6 to the supplemental data, now new Supplemental Fig. 3

Q7. Why the authors do not provide data on the potential bystander edit (A11) in figure 7? The position of the A close to the target site is highly indicative of a potential bystander site. The authors very likely have already the results in their sequence analysis.

R7. We now provide data regarding bystander editing at position 11 in the revised Fig. 6b. With this ABE, we determined that the editing window was from C4 to T8.

Q8. In figure 7 the % of editing referred to a single allele (value of the editing multiplied by 2) should be clearly explained in the text to avoid confusion

R8. We modified the text to clarify as suggested. In the revised Fig. 6 description, the text now states, "... the allelic editing efficiency achieved ... " In addition, we added the following statement to the figure legend and the methods for clarification: "Note that the cells were compound heterozygous for the *CFTR* R553X mutation. Thus, all cells contain "wild type" sequence for one allele and have the R553X mutation on the other allele. This mathematical adjustment provides a meaningful "per cell" estimate of correction." (Main text lines 546-549 and Supplementary Materials lines 292-298).

Q9. in figure 7 the editing efficacy does not correlate with CFTR recovery. The authors should discuss this discrepancy.

R9. There is currently a lack of consensus regarding the % of cells that must be corrected to therapeutically restore CFTR function. Previous studies have indicated that the threshold for complementing CFTR function occurs in cell mixing studies when ~5-10% of cells with wild type CFTR are present within a CF epithelium^{1, 2}. Our results are consistent with these studies. It is likely that restoring CFTR function to certain cell types has greater impact. Recent studies by Okuda and colleagues demonstrated that secretory cells are the dominant airway surface cell type for CFTR expression and function, while ciliated exhibited low and infrequent CFTR expression³. In the same study, secretory cells comprised ~15% of the epithelium. Additionally, recent single cell RNA-seq studies of airway epithelia showed that secretory cells express the Na-K-2Cl cotransporter-1 (NKCC1 or SLC12A2), the basolateral membrane Cl⁻ entry path required for Cl⁻ secretion^{4, 5}. We speculate that because airway epithelial cells are electrically coupled by gap junctions^{6, 7, 8} Cl⁻ may move between secretory cells with no functional CFTR to those that are corrected by gene editing. Thus, any cell with new functional CFTR channels is poised to support Cl⁻ secretion and may provide a conduit through its connections to neighboring cells. We added this information to the revised discussion (Lines 302-327).

Q10. figure 4: why a single inhalation was performed?

R10. The study design was focused on one administration as this is required before any further studies could be considered. There is no rationale to consider multiple rounds without initial findings with one administration that may or may not provide the support to consider additional administration(s).

Q11. Discussion is needed to elaborate on the discrepancy between efficiency of Cy5 and ABE delivery in vivo in rhesus monkey.

R11. This apparent discrepancy is to be expected. Several important differences should be noted when interpreting the delivery efficiency of Cy5-NLS and ABE RNP. First, the physical properties of the two cargoes are quite different, including their molecular weight and charge densities. Second, our assessment of successful delivery varies between the two cargoes.

In response to this query, we added a new paragraph to the discussion: "We observed that NLS-Cy5 delivery was more efficient than ABE8e-Cas9 RNP delivery. There are possibly several reasons for this finding. The NLS-Cy5 is a synthetic peptide of 24 amino acids (VKRKKKPPAAHQSDATAEDDSSYC) with an estimated molecular weight of 2.6 kDa. In contrast, the ABE8e-Cas9 is a much larger protein of 1,614 amino acids and molecular weight of 185 kDa. We previously observed that a smaller cargo (FITC-Dextran 10 kDa) was more efficiently delivered by an amphiphilic peptide than a larger cargo (FITC-Dextran 250 kDa)⁹. Moreover, used as RNP, the guide RNA further increases the size and brings negatively charged nucleic acid moieties to the ABE8e-

Cas9 protein which could reduce S10 peptide delivery activity. Additionally, our assessment of successful delivery varies between the two cargos. While the efficiency of NLS-Cy5 delivered cells is measured directly by visualizing fluorescent epithelial cells, the ABE delivery efficiency is indirectly measured by the gene editing efficiency. Since the editing efficiency is influenced by the individual guide RNA, it remains possible that not all cells that successfully received the ABE RNPs go on to have measurable editing.” Lines 380-393.

Reviewer #2

This is a new manuscript submission by Kulhankova et al. entitled, “Shuttle Peptide Delivers base editor RNPs to Rhesus Monkey airway epithelial cells in vivo”. The work builds on a previous publication identifying shuttle peptides specially the amphiphilic S10 when combined with Cas-RNP can enhance gene editing in human and mouse airway cells. Here, the authors optimized their previously reported S10 shuttle to derive the S315 peptide and used it for adenine base editor RNP (ABE8e-Cas9 RNP) delivery. Using a peptide screen in well differentiated human airway cultures, they identified several new peptides that delivered Cas9 RNP more efficiently than S10, with specific focus on the most highly efficient one S315. Using a rhesus monkey model, intratracheal aerosol delivery of the S10 peptide cargo (Cy5-labelled) resulted in Cy5 fluorescence (peptide distribution) throughout the airways and as deep into the lungs as the alveoli. With this promising result, they then targeted a safe harbor site CCR5 using S315 and show editing efficiency on average ~5% in the airways of these monkeys, with no signs of toxicity or proinflammatory responses. Persistent (up to 12 months) expression of gene editing in the airways was observed using a transgenic mouse model (Ai9 ROSA26 tdTomato), however this was determined using the original S10 shuttle peptide. And finally, codelivery of the S315 and ABE8e-Cas9 RNP in primary human airway epithelial cells heterozygous for the mutation R553X showed restoration of CFTR function and achieved higher levels of editing than the S10 peptide.

Overall, this study builds on their previous work on S10 published in 2019, that now compares S315 with S10 and their ability to improve base editing in relevant animal and human models. There is enthusiasm for this work however, there are some important concerns/clarifications that needs to be addressed.

Q12. How stable is the S315 or any of the shuttle peptides in a pro-inflammatory environment? The study shows great efficiency at targeting cells along the respiratory tree but in the context of disease such as CF, the airways is a highly inflammatory milieu and therefore understanding the stability and targeting efficacy in this context is important especially since the last figure suggests a potential application of this system for CF. One possibility is to test stability in CF patient sputum which is enriched with proteases.

R12. The reviewer makes an important point. In response to this query we evaluated shuttle peptide activity in the presence of sputum from CF patients. While we observed a significant loss of S10 activity in the presence of sputum, the addition of protease

inhibitors rescued shuttle activity, suggesting that the plethora of proteases in the sputum are important factors that inactivate the shuttle peptide. To improve stability in the protease-enriched CF mucus environment, we obtained a D-amino acid version of the S10 peptide, which is resistant to protease degradation. While the S10 D-peptide showed similar delivery activity compared to its L-peptide counterpart in the sputum free condition, the D-peptide remained active in the presence of CF sputum. These results underline the potential applications of the shuttle peptide delivery system in the pro-inflammatory CF lung environment, with the D-peptide as a potential strategy to preserve delivery activity. It is important to note that the L- and D-peptide activity is similar in the absence of sputum *in vitro* or in healthy animals, where proteases do not pose a threat to L-peptide shuttle integrity. We include these new results in a new Supplementary Fig. 7.

We added the following comment to the discussion: “The use of CF animal models that develop pulmonary manifestations may also provide valuable information on shuttle activity in diseased lung tissue. CF patients accumulate abnormal mucus in the airways enriched with mucins, inflammatory cells, proteases, DNA, and bacteria which could interfere with delivery^{10, 11}. Preliminary studies indicate that shuttle peptide delivery activity can be maintained in presence of CF sputum by adding protease inhibitors or using a D-amino acid shuttle peptide (Supplementary Fig. 7a). Since using D-amino acids to synthesize the shuttle does not affect delivery activity *in vitro* and *in vivo* (Supplementary Fig. 7a, b), it represents a simple approach to translate this technology to lung diseases such as CF.” Lines 329-336.

Q13. The rationale for using S10 in the biodistribution studies and not S315 is unclear. The argument is that S315 is a better shuttle peptide as the efficiency of targeting is much higher and therefore, for both biodistribution and toxicity studies, why not evaluate with S315 (irrespective of GFP delivery activity)?

R13. Peptide mediated delivery efficiency will vary according to the cargo. While S315 provides improved delivery of ABE8e-Cas9 RNP, its efficiency for the neutral NLS-Cy5 cargo was less than S10. We include new data in Supplementary Fig. 1b comparing NLS-Cy5 delivery with S315 and S10.

Q14. The authors do note in the discussion that the rhesus monkey experiments only tests 1 bolus (not serial) and 1 concentration (dose) to determine editing efficiency. Thus, the sample size is very small. For biodistribution, using this model is quite relevant to human systems. However, biodistribution here is in the context of a “normal” lung (no inflammation/injury).

- It would be interesting to see if this is also true in an injured setting, and as in the mouse study, whether “persistence” is altered in a setting where there would be more cell turnover (ie. injury).
- Second, it is not clear how the dose was determined and more importantly, how serial dosing or higher concentrations may affect efficacy (albeit improve targeting efficiency) but also toxicity and proinflammatory responses which would be interesting in the primate model. Alternatively, for concerns with costs

associated with the primate model, one possibility it to use the mouse model, perhaps even the humanized CF mouse model?

R14. We agree that these are interesting and important questions. A detailed study of repeated dosing protocols and further investigation into the persistence of edited cells in the setting of injury is a future goal, but outside the scope of the current study. The *in vivo* dose was determined based on our *in vitro* results in well differentiated primary cultures of rhesus and human airway epithelia, and our previous studies in mice¹². We provide new data regarding the *in vivo* toxicity in mice (see Supplementary Figs. 4, 5, 6) and provide additional observations from the rhesus study. We also evaluated the efficacy of editing in the presence of CF sputum, an environment rich in inflammation and proteases, and these data are provided in a new Supplementary Fig. 7 and discussed in lines 329-336. See also the response to reviewer query 3 above.

Q15. The recombination for tdTomato expression appears to be exclusive to the airways (no alveoli) in Fig6, whereas in Fig3, editing was observed in the alveolar regions. Can the authors explain?

R15. Alveolar cells were also edited in the experiment shown in the former Figure 6 (now Suppl. Fig 3), consistent with our previous observation¹² and the results with S10 mediated delivery of NLS-Cy5 in the present study. We provide new images with a more representative picture demonstrating editing within the alveoli. See revised Supplementary Fig. 3.

Q16. In the human CF airway epithelial cell model (Fig 7), while editing efficiency was higher with S315 than S10 (almost 2-fold), functional correction is the same. Can the authors explain why?

Wouldn't S10 be a better candidate peptide for CF R553X base editing in this case? Is the goal here to improve editing efficiency (# of cells targeted) or functional outcome from editing? If the later, S10 seems to be just as good at delivering base editors in this case. It was not clear how many patient cell lines were used here? There are many primary cell biobanks to test additional lines and potentially other mutations.

R16. As presented in Fig. 2, we found that for ABE8e-Cas9 RNP, the delivery efficiency was greater with S315 than S10. Our gene editing results presented in Fig. 6c (formerly Fig. 7C) also support this conclusion.

The results presented in the revised Fig. 6 are from cells from one donor with the R553X mutation on one allele. For both peptides, the editing achieved is at levels believed to be near the threshold for detecting CFTR channel activity by short circuit current analysis. In addition, as now discussed in more detail in the revised discussion (Lines 302-327), the editing outcomes as assessed by CFTR-dependent Cl⁻ secretion will likely depend on which cell types are edited. For example, secretory cells express more CFTR than ciliated cells and the relative proportions of the edited surface cell types is expected to influence the functional outcomes.

Q17. It is unclear if there were off-target effects with improved editing efficiency. Please clarify.

R17. We anticipate that if more ABE8e-Cas9 RNP is delivered to cells, we will see an increase in the editing efficiency and may also observe an increase in bystander editing. In the revised Fig. 6b we present additional bystander editing results for A11. We observed a small increase in the frequency of editing at A11 for S315 (1.03%) vs S10 (0.53%). Note that an A>G edit at A11 would introduce a synonymous mutation and retain the Gly codon. We also added bystander editing information to Figs. 2a,c.

Reviewer #3

The manuscript by Kulhankova and colleagues reports an array of experiments validating a pre-clinical strategy – pairing a base editor enzyme with a novel delivery approach – that has translational potential for therapeutic correction of cystic fibrosis (CF). Although small molecule drugs can currently benefit CF patients with certain mutations, there are still many who harbor mutations that underlie a form of CF that cannot be addressed with existing drugs. One such mutation is R553X, which is the focus of the present work. Prior work by the Liu/McCray team has demonstrated that an adenine base editor (ABE) can enact a genome-level correction for this mutation that would be anticipated to provide clinical benefit in any tissue receiving such a correction. However, delivery of genome editing enzymes to the lung epithelium has not yet been attained in the clinic. Although some pre-clinical work suggests lipid nanoparticles (bearing mRNA or RNP cargo) may hold promise for therapeutic editing of the lung epithelium, delivery to the lung is by no means a solved problem. To address this unmet need, Kulhankova and colleagues have applied (and improved upon, via novel peptide discovery) a strategy previously established by the Guay/McCray team: mixing pre-formed ABE enzymes with amphiphilic peptides that facilitate productive intracellular delivery of RNP enzyme cargo.

In the present manuscript, these two key technologies (therapeutic base editing and peptide-mediated enzyme delivery) are shown to work well together in multiple contexts: human and primate tissue culture models of airway epithelium, the same model using CF-patient derived cells, as well as in vivo administration for editing in the lung epithelium in primates. Additional evidence for the translational approach is provided by complementary experiments: assessment of the cell types being reached using their delivery approach as well as an assay evaluating the persistence of edited cells over the course of a year in a fluorescent reporter mouse model. The study concludes with a thoughtful and precise analysis of the likely clinical endpoints pertinent for development a functional CF therapy. Encouragingly, this analysis accurately finds that the present pre-clinical approach holds substantial promise for successful clinical translation. The authors would be well justified to express even more optimism regarding their capacity to advance the field of genetic therapy for CF: a virally-vectored traditional gene therapy approach has apparently languished for years, indicating the need for new and innovative approaches to address the patients who cannot benefit from current CF drugs.

I recommend publication of this manuscript after a two minor points are suitably addressed. As is described in detail below, I have doubts regarding certainty of the conclusions associated with the GFP/Cas9 “competition” assay reported in Fig. 1c. Also described in detail below is a concern regarding the processing/reporting of sequencing data is discussed. With these two issues having been corrected or clarified, I believe this manuscript will be an outstanding candidate for publication by Nature Communications. I have also provided feedback that strives to strengthen the manuscript, but these additional points may be prioritized according to the authors’ and editor’s discretion.

Below, main-text line numbers will be marked as M####; supplementary information line numbers will be marked as S###

Major concerns

Q18. M109 / Fig. 1c / Supp. Fig. 1 (experiment & conclusions)

I am not sure these experimental results can only be explained via the proposed mechanism. The authors suggest that diminished GFP delivery (by peptide S10) is due to the peptide’s “susceptibility to inhibition by Cas9 RNP”, and this finding is extended (if I understand correctly) to insinuate that formulation containing S10 and Cas9 would somehow be harmful for delivery of Cas9. But previous work has clearly shown that S10 is perfectly capable of facilitating Cas9 delivery both in tissue culture models and in vivo. The assay reported in Fig. 1c is most directly assessing each peptide’s ability to deliver GFP, and how that ability is impacted by the presence of Cas9 RNP. I don’t think these results can rule out a model wherein S10 peptide favors non-covalent association with Cas9 RNP (as opposed to GFP). In contrast, peptide S315 could favor interactions with GFP (along with, or instead of with Cas9 RNP). This is an alternative mechanism/model that explains the results shown in Fig. 1c, and it doesn’t have particular explanatory value for the manuscript writ large. I suggest the authors either remove their narrow mechanistic interpretation of the experiment, augment their explanation to include the alternative model, or perform an additional experiment to narrow the possibilities. Regarding the latter, one such experiment could be a Cas9 RNP-mediated genome editing assay that is performed using either peptide, in the absence or presence of GFP. If my proposed model is correct, editing levels will not be impacted by the presence of GFP when S10 is used, while S315-mediated editing will be constant or diminished in the presence of GFP. To be clear, I don’t think these experiments – that of Fig. 1c and/or my proposed experiment – are essential for the manuscript’s overall conclusions. Mechanistic understanding of delivery is valuable but not necessary.

R18. Thank you for this comment. We deleted the sentence referring to a possible mechanism for this finding. See Lines 289 – 300.

Q19. S249-250 (and legend of Fig. 7)

The supplement states: “The allelic editing frequency was calculated by the following equation: $(\% \text{ editing} - 250) / 50$.” Make sure this equation is reported consistently: the

supplement matches text in Fig. 7, but the Fig. 7 legend diverges in that it has “* 100” appended. I don’t think this is likely to cause substantial confusion, but it’s nice to be consistent. In contrast, I think the typical reader may actually be confused about how/why 52.5% editing represents ~5% editing. It would be helpful to add a brief clarifying statement in the Results and/or Methods sections noting that all cells contain “wild type” sequence (at the genomic locus being assessed) for one allele, and bear the R553X mutation on the other allele – hence the mathematical adjustment, which provides a meaningful “per cell” estimate of correction.

R19. Thank you for this suggestion. We added the following to the legend of Fig. 6 (former Fig. 7) and to the Supplementary Methods section: “Note that for the results obtained for the human R553X mutation, the cells were compound heterozygous for this *CFTR* mutation. Therefore, all cells contain “wild type” sequence (at the genomic locus being assessed) for one allele and have the R553X mutation on the other allele. This mathematical adjustment provides a meaningful “per cell” estimate of correction.” See Main text lines 546-549 and Supplementary Information lines 292-298.

Suggestions

Q20. Discussion: It might be valuable to highlight the potential strengths of this approach as it is moved towards the clinic. In addition to highlighting the strengths of non-viral CRISPR delivery (for decreased immunogenicity and genotoxicity), there may be advantages in terms of GMP & CMC. GMP CRISPR RNP found its way to clinical use extremely rapidly (CRISPR Therapeutics’ SCD trial) and peptide should be quite amenable to GMP manufacture. This is a major real-world advantage in contrast to viral manufacture, which can be quite complicated and prone to complications & delays.

R20. We added the following sentence to the discussion: “Peptide and base editor proteins should also be amenable to rapid Good Manufacturing Practices (GMP) production and chemistry, manufacturing, and control (CMC) testing.” See lines 354-356.

Q21. M63 Regarding lipid-based delivery of RNP to the lung epithelium, perhaps Cheng et al. 2020 (<https://www.ncbi.nlm.nih.gov/pmc/articles/PMC7735425/>) should be considered. RNP delivery to the lung is reported in Fig. 4f, and the microscopy might be sufficiently compelling to warrant mention here.

R21. Thank you for this suggestion. The following sentence was edited in the introduction as follows: “Several delivery strategies including viral and non-viral vectors are in development to enable the delivery of editing reagents to somatic cells *in vivo*, including the lung^{13, 14, 15, 16, 17, 18}.” Lines 60-62.

Q22. M67 Consider revision to “avenues for Cas nuclease and base editor RNP delivery”; “Cas” is essentially an adjective, and “nuclease” is the appropriate term to contrast against “base editor”.

R22. We modified this sentence as suggested. Line 70.

Q23. M94 Do you really intend to describe lysine as a “di-basic” residue? I believe it should be described as “basic”.

R23. We changed “di-basic” to “basic” in these sentences and in Fig. 1b. Lines 95 and 99.

Q24. M121 Consider revision to “greater than with S10”

R24. We revised the sentence as suggested. See line 126-7.

Q25. M144 Consider revising to “(Fig. 3b, top)”

R25. We revised as suggested. See line 149.

Q26. M149 Consider revising to “(Fig. 3b, middle)”

R26. We revised as suggested. See line 154.

Q27. M152 Consider revising to “(Fig. 3b, bottom)”

R27. We revised as suggested. See line 157.

Q28. M168 / M188 / S153 It’s difficult to assess how much material is being administered because the relevant values are not reported together in the same place. For example, M168 reports concentrations, but the volume is only reported at S153.

R28. We added the volume administered with the concentration in the Results Section as follows: “Using an approach identical to that outlined for the NLS-Cy5 peptide, we delivered 1 ml of instillation solution containing ABE8e-Cas9 RNP (2.5 μ M Cas/2 μ M gRNA final concentrations; 40 μ M Cas/100 μ M crRNA and tracrRNA stock concentrations) formulated with either the S10 or S315 peptide (40 μ M final concentration; 250 μ M stock concentration) in DPBS by intratracheal aerosol as described in Methods.” See lines 171-176

Q29. M191 For readers who are not pulmonologists, it would be helpful to include a very brief explanation for the role of ground glass in CT scans. Furthermore, it’s probably worth presenting CT in its expanded form (computed tomography) the first time it is introduced.

R29. We expanded the word CT as “computed tomography (CT)”. see Line 212. We also added the following sentence regarding the CT scoring. “A pulmonologist blinded to the experimental conditions then scored the CT scans for the presence and extent of parenchymal lung changes indicated by areas of abnormal density.” See lines 217-219.

Q30. M215 Consider revision to “There was a statistically significant decline in the number of tdTomato+ cells from 7 days to 12 months, but no such decline was apparent when comparing the 7 day time point to other time points up to 6 months.”

R30. The sentence was revised as suggested. See lines 242-244.

Q31. M222 Consider citing Krishnamurthy 2021, since it provides helpful context regarding the use of the NG PAM for R553X base editing.

R31. The reference was added as suggested. See line 262.

Q32. M250-251 Consider revision to “The present demonstration”

R32. The sentence was revised as suggested. See line 291.

Q33. M271-272 Consider revision to “While ionocytes express the highest levels of CFTR transcripts, they are a rare cell type and their function is still a subject of study, as is their role in genetic therapies for CF.”

R33. The sentence was revised as suggested. See lines 309-311.

Q34. M279-280 Consider rephrasing “targeted cell types” – this seems to imply deliberate targeting of particular cell types, which isn’t a focus of the approaches employed by this study. Perhaps “preferentially-edited cell types” would be more appropriate.

R34. The statement was revised as suggested. See line 339.

Q35. M287 Consider revision to “The strategy presented here has strengths and limitations.” The current wording – “This study has advantages” – seems to reflect the study design, not the pre-clinical therapeutic strategy.

R35. The sentence was revised as suggested. See line 347.

Q36. M303 Consider revision to “Our results in Ai9 mice following editing using MAD7 nuclease”

R36. The sentence was revised as suggested. See line 372.

Q37. Fig. 1b For the boxes around basic residues, consider a color pair other than red & green, since it’s not colorblind friendly. In this case, red & black should suffice. For the arrow noting the transplanted leucine, the arrowhead should be pointing at the red leucine, not the black leucine marked as “conserved”.

R37. Figure 1b was revised to indicate the modified leucine as suggested.

Q38. Fig. 4c Consider adding “CCR5” to this panel, preferably as part of the y-axis.

R38. We revised Fig. 4c as suggested.

Q39. Fig. 7c As above, consider adding “R553X locus” to y-axis.

R39. We revised Fig. 6c (formerly Fig. 7c) as suggested.

Q40. S144 Although sterile conditions are noted for various steps, there is no mention of sterile filtration of the CRISPR protein(s). If the recombinant proteins (produced in *E. coli*) are not sterile, it is unclear how much the downstream steps matter.

R40. We agree with the comment and confirm that all the CRISPR proteins are filter sterilized using a 0.2 μ M filter. For clarity, we modified the section “Expression and purification of recombinant ABE8e protein and MAD7 nuclease” in the Supplementary Information to include this information. The modified sentence is copied here with the modifications highlighted in bold fonts for the reviewer: “The peak corresponding to ABE8e-Cas9 protein dimers was concentrated on AMICON 100K to a final concentration of 40 μ M, **filter sterilized using a 0.2 μ m filter, aliquoted under the laminar flow hood**, snap-frozen in liquid nitrogen, and stored at -80°C. The final product had an endotoxin concentration of <0.250 EU/ μ g.” See lines 64-68 of Supplementary Information.

Q41. S147 “Procedures” heading is too broad to be useful; revise to a more specific heading, such as “Rhesus macaque in vivo procedures”.

R41. We revised the headings to a more specific description of the procedures.

Reviewer #4 (Remarks to the Author):

Comments to authors

This is a very well-written manuscript, the figures are clearly presented and the study represents an advance in the field. Strengths include the use of nonhuman primates to extend from in vitro and mouse studies, which provides the potential for further development. Weaknesses include uncertainty if the 5% transduction and editing efficiency demonstrated with the monkey study will be long-lived, or provide a physiological amelioration of the CF condition. The study would benefit from monkey trials that last beyond 1 week.

Specific comments

Abstract

Q42. Line 29: 5.3% is at the very low end of the therapeutically beneficial range cited later in the ms. While persistence was documented in mice, the manuscript would be far stronger if monkey studies were carried out beyond 1 week.

R42. For this initial study, the goal was to address short-term (1 week) outcomes. Future studies will focus on long-term findings and based on the outcomes of this short-term initial study for the experimental plan. Please see response to comment 49 below.

Q43. Line 114: it is not clearly explained why the B2M locus was targeted for base editing.

R43. The *B2M* locus was selected because it is a ubiquitously expressed gene product in diverse cell types including airway epithelia. We added this statement to the results section. Lines 119-120.

Q44. Line 121: a 9% *in vitro* efficiency is difficult to translate to a much less favorable *in vivo* environment.

R44. Figure 2b validated the function of the guide RNA target *CCR5* locus with rhesus monkey airway cells cultured at air liquid interface (ALI), and confirmed previous observation that S315 is better than S10 at delivering base editors. Although 9% *in vitro* efficiency is relatively low, it is an important step towards the subsequent *in vivo* study which achieved a up to 5.3% base editing efficiency with the same base editor and guide RNA. We provide new results regarding peptide mediated delivery in the presence of protease rich CF sputum (Supplementary Fig. 7).

We note that if an effective delivery strategy is developed and proven safe, a long-term goal would be to administer it early in life to a more pristine airway environment.

Q45. Line 144: it would be helpful to provide additional letter labels to the panels of Figure 3b, it is not immediately obvious which panel of the figure is being referred to since there are multiple components of Fig 3b.

R45. As outlined in our responses to Q25-27 above, we edited the text to draw the reader's attention to the appropriate panels.

Q46. Lines 163-4. There is a very wide range of labeling efficiencies. This was not discussed as a limitation of the study but is it significant.

R46. In our experience, a single intratracheal aerosol bolus administration will often result in heterogeneous deposition of reagents within the airways and alveoli. Some areas will receive more reagent than others; some regions may receive little or no reagent. We and others have noted this with other delivery strategies in animal models and do not find this result surprising. The point of including the CT scans was to provide insight on areas of deposition. A future goal will be to optimize an efficient delivery approach to ensure more uniform delivery which in future rhesus studies will be confirmed with CT scans. We added the following statement to the discussion: "We note that a single intratracheal aerosol delivery resulted in heterogeneous deposition of the reagents. A future goal will be to consider ways to optimize more uniform delivery." See

lines 366-368.

Q47. Lines 213 and 215 seem to convey contradictory messages. 215 says tdTomato was persistent for 12 months but lines 214-5 say that there was a decline in the number.

R47. We added the following sentence for clarification: "There was a statistically significant decline in the number of tdTomato⁺ cells from 7 days to 12 months, but no such decline was apparent when comparing the 7-day time point to other time points up to 6 months." See lines 242-244.

Q48. Lines 234-6. While Fig 7 shows cellular impact, the study would be strengthened by demonstration of a physiological therapeutic impact on animal well-being.

R48. There is currently no non-human primate model of cystic fibrosis available. As noted in the text, the overall status of the animals was maintained for the duration of the study. For those in the 7-day investigation there was no evidence of adverse findings once the animals were fully awake and returned to the housing area. They resumed normal food intake and activities as expected for animals in this age group. See also comments in R3 above. See lines 177-183.

Discussion

Q49. Line 262: given that the therapeutic range stated is 5-50% restoration needed, it is important to note that the short-term monkey study was at the low end of this range.

R49. We acknowledge this point. We revised the discussion to state: "While this editing efficiency is at the low end of the therapeutic range, application of this delivery approach in human CF airway epithelia with the R553X mutation achieved similar levels of editing and conferred partial restoration of CFTR function." Also, see response to Query 9 from Reviewer 1. Lines 284-287.

Q50. Line 281: for me it wasn't clearly explained why CCR5 was targeted.

R50. To clarify our choice of *CCR5* gene, we edited the sentence as follows, and included the respective reference: "While CCR5 is not an abundant transcript in airway epithelia¹⁹, it has been studied extensively as a genomic safe harbor site for gene therapies²⁰, and its low-level expression suggests accessible open chromatin." We also considered that making an edit in a safe harbor locus was unlikely to cause harm to an otherwise healthy animal, which was shown to be the case. Lines 340-342.

References cited

1. Johnson LG, Olsen JC, Sarkadi B, Moore KL, Swanstrom R, Boucher RC. Efficiency of gene transfer for restoration of normal airway epithelial function in cystic fibrosis. *Nat Genet* **2**, 21-25 (1992).

2. Dannhoffer L, Blouquit-Laye S, Regnier A, Chinet T. Functional properties of mixed cystic fibrosis and normal bronchial epithelial cell cultures. *Amer J Resp Cell Molec Biol* **40**, 717-723 (2009).
3. Okuda K, *et al.* Secretory Cells Dominate Airway CFTR Expression and Function in Human Airway Superficial Epithelia. *Amer J Resp Crit Care Med*, (2020).
4. Montoro DT, *et al.* A revised airway epithelial hierarchy includes CFTR-expressing ionocytes. *Nature* **560**, 319-324 (2018).
5. Plasschaert LW, *et al.* A single-cell atlas of the airway epithelium reveals the CFTR-rich pulmonary ionocyte. *Nature* **560**, 377-381 (2018).
6. Wiszniewski L, *et al.* Functional expression of connexin30 and connexin31 in the polarized human airway epithelium. *Differentiation* **75**, 382-392 (2007).
7. Scheckenbach KE, *et al.* Prostaglandin E(2)regulation of cystic fibrosis transmembrane conductance regulator activity and airway surface liquid volume requires gap junctional communication. *Amer J Resp Cell Molec Biol* **44**, 74-82 (2011).
8. Boitano S, Dirksen ER, Sanderson MJ. Intercellular propagation of calcium waves mediated by inositol trisphosphate. *Science* **258**, 292-295 (1992).
9. Del'Guidice T, *et al.* Membrane permeabilizing amphiphilic peptide delivers recombinant transcription factor and CRISPR-Cas9/Cpf1 ribonucleoproteins in hard-to-modify cells. *PLoS One* **13**, e0195558 (2018).
10. Fahy JV, Dickey BF. Airway mucus function and dysfunction. *N Engl J Med* **363**, 2233-2247 (2010).
11. Rubin BK. Mucus, phlegm, and sputum in cystic fibrosis. *Resp Care* **54**, 726-732; discussion 732 (2009).
12. Krishnamurthy S, *et al.* Engineered amphiphilic peptides enable delivery of proteins and CRISPR-associated nucleases to airway epithelia. *Nat Comm* **10**, 4906 (2019).
13. Excoffon KJ, *et al.* Directed evolution of adeno-associated virus to an infectious respiratory virus. *PNAS* **106**, 3865-3870 (2009).
14. Wei T, Cheng Q, Min YL, Olson EN, Siegwart DJ. Systemic nanoparticle delivery of CRISPR-Cas9 ribonucleoproteins for effective tissue specific genome editing. *Nat Comm* **11**, 3232 (2020).
15. Sago CD, *et al.* Augmented lipid-nanoparticle-mediated in vivo genome editing in the lungs and spleen by disrupting Cas9 activity in the liver. *Nat Biomed Eng* **6**, 157-167 (2022).
16. Banskota S, *et al.* Engineered virus-like particles for efficient in vivo delivery of therapeutic proteins. *Cell* **185**, 250-265 e216 (2022).

17. Liang SQ, *et al.* AAV5 delivery of CRISPR-Cas9 supports effective genome editing in mouse lung airway. *Molec Ther* **30**, 238-243 (2022).
18. Cheng Q, Wei T, Farbiak L, Johnson LT, Dilliard SA, Siegwart DJ. Selective organ targeting (SORT) nanoparticles for tissue-specific mRNA delivery and CRISPR-Cas gene editing. *Nat Nanotechnol* **15**, 313-320 (2020).
19. Chinnapaiyan S, *et al.* Cigarette smoke promotes HIV infection of primary bronchial epithelium and additively suppresses CFTR function. *Sci Rep* **8**, 7984 (2018).
20. Aznauryan E, *et al.* Discovery and validation of human genomic safe harbor sites for gene and cell therapies. *Cell Rep Methods* **2**, 100154 (2022).

Reviewers' Comments:

Reviewer #1:

Remarks to the Author:

The authors addressed all concerns from the revision. This is a valuable study advancing the field of gene therapy for lung diseases. Publication is recommended.

Reviewer #2:

Remarks to the Author:

This is a significantly improved revision of the manuscript. The authors have addressed many of the reviewers' concerns/questions and detailed these changes in the revised manuscript, revised some of the main manuscript figures and supplementary file to include additional experiments. I only have 1 minor comment in line 242 "There was a statistically significant decline in the number of tdTomato+ cells from 7 days to 12 months, but no such decline was apparent when comparing the 7-day time point to other time points up to 6 months." The sentence is confusing. If there was a decline in number of TdTomato cells from 7 days to 12 months, then why is no such decline apparent when comparing 7 days to any time points up to 6 months? Consider revising this sentence. I'm not sure what the second part means exactly or if it is necessary. It is expected to decline over time especially with cell turnover and if the delivery mainly targets surface cells, you would expect the targeted cells to slough off over time.

As I previously mentioned, this work builds on a previous work with S10 peptide. The team has now improved the shuttle peptide S315 which shows much better efficiency at delivering base editor RNP. They test this in cells, mouse models (for long-term persistence evaluation) and rhesus macaques (for preclinical relevance). Overall, I am happy with the revisions and recommend acceptance of the paper.

Reviewer #3:

Remarks to the Author:

The revised manuscript addresses the concerns I raised earlier. Furthermore, I believe the other reviewer's questions and concerns have been addressed sufficiently. I support publication of this manuscript.

REVIEWERS' COMMENTS

Reviewer #2 (Remarks to the Author):

Q1.

This is a significantly improved revision of the manuscript. The authors have addressed many of the reviewers' concerns/questions and detailed these changes in the revised manuscript, revised some of the main manuscript figures and supplementary file to include additional experiments. I only have 1 minor comment in line 242 "There was a statistically significant decline in the number of tdTomato+ cells from 7 days to 12 months, but no such decline was apparent when comparing the 7-day time point to other time points up to 6 months." The sentence is confusing. If there was a decline in number of TdTomato cells from 7 days to 12 months, then why is no such decline apparent when comparing 7 days to any time points up to 6 months? Consider revising this sentence. I'm not sure what the second part means exactly or if it is necessary. It is expected to decline over time especially with cell turnover and if the delivery mainly targets surface cells, you would expect the targeted cells to slough off over time.

As I previously mentioned, this work builds on a previous work with S10 peptide. The team has now improved the shuttle peptide S315 which shows much better efficiency at delivering base editor RNP. They test this in cells, mouse models (for long-term persistence evaluation) and rhesus macaques (for preclinical relevance). Overall, I am happy with the revisions and recommend acceptance of the paper.

R1.

Thank you for this suggestion. We modified the sentence as follows: "There was a statistically significant decline in the number of tdTomato+ cells from 7 days to 12 months."